# Population clustering of structural brain aging and its association with brain development

Haojing Duan[1,2], Runye Shi[3], Jujiao Kang[1,2], Tobias Banaschewski[4], Arun LW Bokde[5], Christian Büchel[6], Sylvane Desrivières[7], Herta Flor[8,9], Antoine Grigis[10], Hugh Garavan[11], Penny A Gowland[12], Andreas Heinz[13], Rüdiger Brühl[14], Jean-Luc Martinot[15], Marie-Laure Paillère Martinot[15,16], Eric Artiges[15,17], Frauke Nees[4,8,18], Dimitri Papadopoulos Orfanos[10], Luise Poustka[19], Sarah Hohmann[4], Nathalie Nathalie Holz[4], Juliane Fröhner[20], Michael N Smolka[20], Nilakshi Vaidya[21], Henrik Walter[13], Robert Whelan[22], Gunter Schumann[1,21,23,24], Xiaolei Lin[3,25]*, Jianfeng Feng[1,2,3,23,26,27,28]*

[1]Institute of Science and Technology for Brain-Inspired Intelligence, Fudan University, Shanghai, China; [2]Key Laboratory of Computational Neuroscience and Brain-Inspired Intelligence (Fudan University), Ministry of Education, Shanghai, China; [3]School of Data Science, Fudan University, Shanghai, China; [4]Department of Child and Adolescent Psychiatry and Psychotherapy, Central Institute of Mental Health, Medical Faculty Mannheim, Heidelberg University, Mannheim, Germany; [5]Discipline of Psychiatry, School of Medicine and Trinity College Institute of Neuroscience, Trinity College Dublin, Dublin, Ireland; [6]University Medical Centre Hamburg-Eppendorf, Hamburg, Germany; [7]Social Genetic and Developmental Psychiatry Centre, Institute of Psychiatry, Psychology and Neuroscience, King's College London, London, United Kingdom; [8]Institute of Cognitive and Clinical Neuroscience, Central Institute of Mental Health, Medical Faculty Mannheim, Heidelberg University, Mannheim, Germany; [9]Department of Psychology, School of Social Sciences, University of Mannheim, Mannheim, Germany; [10]NeuroSpin, CEA, Université Paris-Saclay, Gif-sur-Yvette, France; [11]Departments of Psychiatry and Psychology, University of Vermont, Burlington, United States; [12]Sir Peter Mansfield Imaging Centre School of Physics and Astronomy, University of Nottingham, Nottingham, United Kingdom; [13]Department of Psychiatry and Psychotherapy CCM, Charité – Universitätsmedizin Berlin, corporate member of Freie Universität Berlin, Humboldt-Universität zu Berlin, and Berlin Institute of Health, Berlin, Germany; [14]Physikalisch-Technische Bundesanstalt (PTB), Braunschweig and Berlin, Berlin, Germany; [15]Institut National de la Santé et de la Recherche Médicale, INSERM U1299 "Developmental Trajectories and Psychiatry", Université Paris-Saclay, Ecole Normale supérieure Paris-Saclay, CNRS, Centre Borelli, Gif-sur-Yvette, France; [16]AP-HP. Sorbonne Université, Department of Child and Adolescent Psychiatry, Pitié-Salpêtrière Hospital, Paris, France; [17]Psychiatry Department, EPS Barthélémy Durand, Etampes, France; [18]Institute of Medical Psychology and Medical Sociology, University Medical Center Schleswig-Holstein, Kiel University, Kiel, Germany; [19]Department of Child and Adolescent Psychiatry and Psychotherapy, University Medical Centre, Göttingen, Germany; [20]Department of Psychiatry and Neuroimaging Center, Technische Universität Dresden, Dresden, Germany; [21]Department of Psychiatry and Neurosciences, Charité–Universitätsmedizin Berlin, corporate member of Freie Universität BerlinHumboldt-

*For correspondence:
xiaoleilin@fudan.edu.cn (XL);
jianfeng64@gmail.com (JF)

Universität zu Berlin, and Berlin Institute of Health, Berlin, Germany; [22]School of Psychology and Global Brain Health Institute, Trinity College Dublin, Dublin, Ireland; [23]Centre for Population Neuroscience and Stratified Medicine (PONS Centre), ISTBI, Fudan University, Shanghai, China; [24]Centre for Population Neuroscience and Stratified Medicine (PONS), Department of Psychiatry and Psychotherapy, Charité Universitätsmedizin, Berlin, Germany; [25]Huashan Institute of Medicine, Huashan Hospital affiliated to Fudan University, Shanghai, China; [26]MOE Frontiers Center for Brain Science, Fudan University, Shanghai, China; [27]Zhangjiang Fudan International Innovation Center, Shanghai, China; [28]Department of Computer Science, University of Warwick, Warwick, United Kingdom

**Abstract** Structural brain aging has demonstrated strong inter-individual heterogeneity and mirroring patterns with brain development. However, due to the lack of large-scale longitudinal neuroimaging studies, most of the existing research focused on the cross-sectional changes of brain aging. In this investigation, we present a data-driven approach that incorporate both cross-sectional changes and longitudinal trajectories of structural brain aging and identified two brain aging patterns among 37,013 healthy participants from UK Biobank. Participants with accelerated brain aging also demonstrated accelerated biological aging, cognitive decline and increased genetic susceptibilities to major neuropsychiatric disorders. Further, by integrating longitudinal neuroimaging studies from a multi-center adolescent cohort, we validated the 'last in, first out' mirroring hypothesis and identified brain regions with manifested mirroring patterns between brain aging and brain development. Genomic analyses revealed risk loci and genes contributing to accelerated brain aging and delayed brain development, providing molecular basis for elucidating the biological mechanisms underlying brain aging and related disorders.

## eLife assessment

Duan et al analyzed brain imaging data in UKBK and divided structural brain aging into two groups, revealing that one group is more vulnerable to aging and brain-related diseases compared to the other group. Such subtyping could be **valuable** and utilized in predicting and diagnosing cognitive decline and neurodegenerative brain disorders in the future. This discovery, supported by **solid** evidence, harbors a substantial impacts in aging and brain structure and function.

## Introduction

The structure of the brain undergoes continual changes throughout the entire lifespan, with structural brain alterations intimately linking brain development and brain aging (*Fjell and Walhovd, 2010*; *Shaw et al., 2008*). Brain aging is a progressive process that often co-occurs with biological aging and declines of cognitive functions (*Elliott et al., 2021*; *Mattson and Arumugam, 2018*; *Park and Reuter-Lorenz, 2009*), which contribute to the onset and acceleration of neurodegenerative (*Mariani et al., 2005*) and neuropsychiatric disorders (*Kaufmann et al., 2019*). Studies on healthy brain aging have revealed significant inter-individual heterogeneity in the patterns of neuroanatomical changes (*Raz et al., 2010*; *Raz and Rodrigue, 2006*). Therefore, examining the patterns of structural brain aging and its associations with cognitive decline is of paramount importance in understanding the diverse biological mechanisms of age-related neuropsychiatric disorders.

Despite the fact that there exist large differences between brain development and brain aging (*Courchesne et al., 2000*), a discernible association between these two processes remains evident. Direct comparisons of brain development and brain aging using structural MRI indicated a 'last in, first out' mirroring pattern, where brain regions develop relatively late during adolescence demonstrated accelerated degeneration in older ages (*McGinnis et al., 2011*; *Tamnes et al., 2013*). In addition, brain regions with strong mirroring effects showed increased vulnerability to neurodegenerative and neuropsychiatric disorders, including Alzheimer's disease and schizophrenia (*Douaud et al., 2014*).

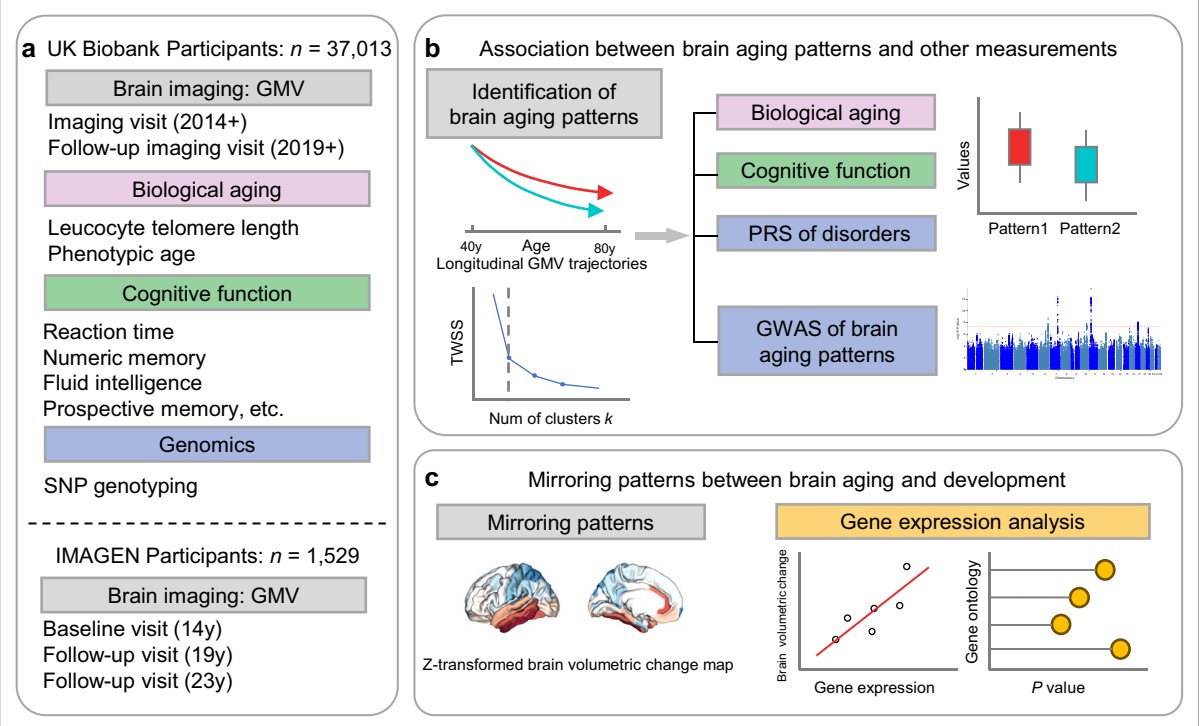

**Figure 1.** Overview of the study workflow. (**a**) Population cohorts (UK Biobank and IMAGEN) and data sources (brain imaging, biological aging biomarkers, cognitive functions, genomic data) involved in this study. (**b**) Brain aging patterns were identified using longitudinal trajectories of the whole brain GMV, which enabled the capturing of long-term and individualized variations compared to only use cross-sectional data, and associations between brain aging patterns and other measurements (biological aging, cognitive functions and PRS of major neuropsychiatric disorders) were investigated. (c) Mirroring patterns between brain aging and brain development was investigated using z-transformed brain volumetric change map and gene expression analysis.

However, due to the lack of large-scale longitudinal MRI studies during adolescence and mid-to-late adulthood, validation of the 'last in, first out' mirroring hypothesis remains unavailable.

Prior investigations have largely focused on regional and cross-sectional changes of brain aging (*Raz and Rodrigue, 2006*; *Douaud et al., 2014*; *Suzuki et al., 2019*), with relatively few studies exploring longitudinal trajectories of brain aging and its associations with brain development (*Raz et al., 2010*; *Fjell et al., 2015*; *Nyberg et al., 2023*). In this article, we present a data-driven approach to examine the population clustering of longitudinal brain aging trajectories using structure MRI data obtained from 37,013 healthy individuals during mid-to-late adulthood (44–82 years), and explore its association with biological aging, cognitive decline and susceptibilities for neuropsychiatric disorders. Further, mirroring patterns between longitudinal brain development and brain aging are investigated by comparing the region-specific aging / developmental trajectories, and manifestation of the mirroring patterns are investigated across the whole-brain and among participants with different brain aging patterns. Genomic analyses are conducted to reveal risk loci and genes associated with accelerated brain aging and delayed brain development.

## Results

### Longitudinal trajectories of whole-brain gray matter volume in mid-to-late adulthood define two brain aging patterns

*Figure 1* provides the data sources, analytical workflow and research methodology of this study. After the sample selection process (*Appendix 1—figure 1*, *Appendix 1—table 1 and 2*), longitudinal grey matter volume (GMV) trajectories in 40 ROIs (33 cortical and 7 subcortical ROIs, see *Appendix 1—table 3*) were estimated for each of the 37,013 healthy participants in UK Biobank. The first 15 principal components derived from dimensionality reduction via principal component analysis were used

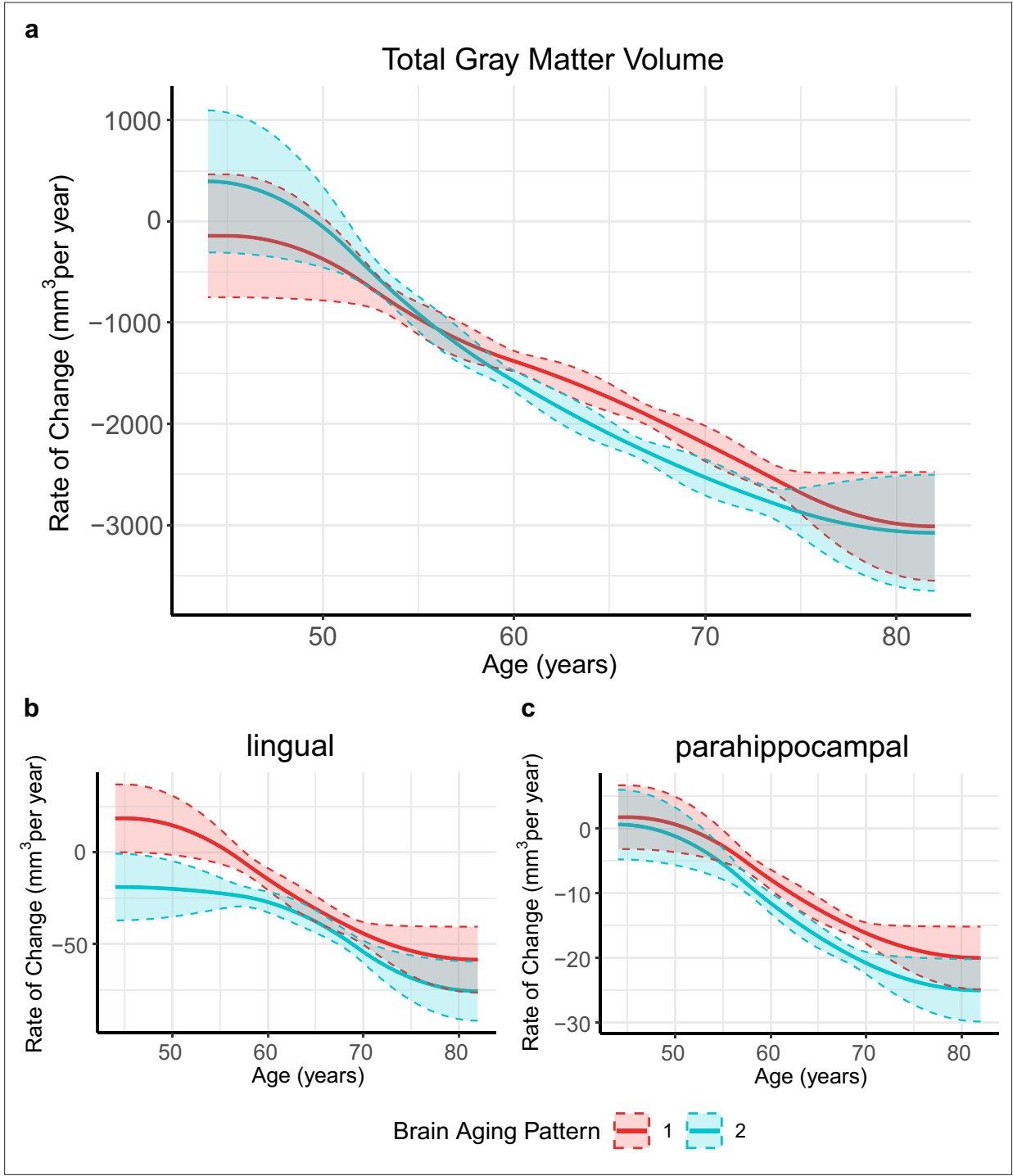

**Figure 2.** Global (**a**) and selected regional (**b, c**) cortical gray matter volume rate of change among participants with brain aging patterns 1 (red) and 2 (blue). Rates of volumetric change for total gray matter and each ROI were estimated using GAMM, which incorporates both cross-sectional between-subject variation and longitudinal within-subject variation from 40,921 observations and 37,013 participants. Covariates include sex, assessment center, handedness, ethnic, and ICV. Shaded areas around the fit line denotes 95% CI.

The online version of this article includes the following source data for figure 2:

**Source data 1.** Related to *Figure 2*.

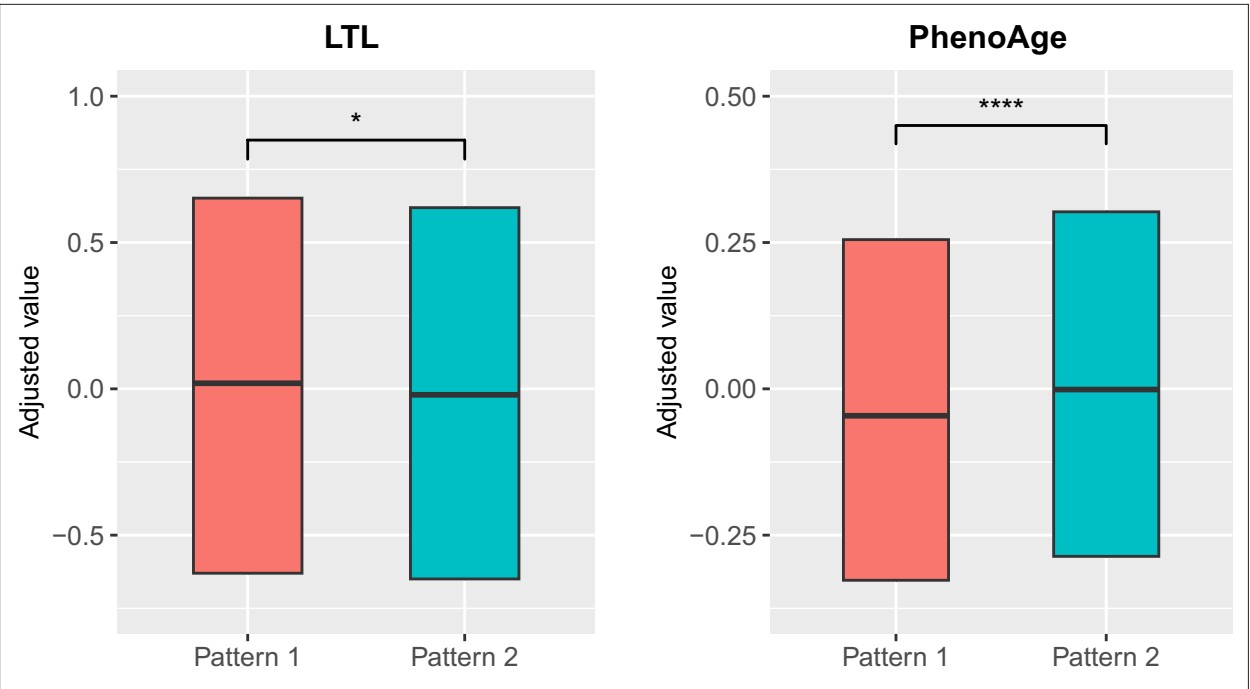

**Figure 3.** Distributions of biological aging biomarkers (leucocyte telomere length (LTL) and PhenoAge) among participants with brain aging patterns 1 and 2. Boxes represent the interquartile range (IQR), lines within the boxes indicate the median. Two-sided p values were obtained by comparing LTL or PhenoAge *Levine et al., 2018* between brain aging patterns using unadjusted multivariate linear regression models. Results remained significant when adjusting for sex, age, ethnic, BMI, smoking status and alcohol intake frequency in the LTL model *Demanelis et al., 2020* and sex, age, ethnic, BMI, smoking status, alcohol frequency and education years in PhenoAge model. Stars indicate statistical significance after Bonferroni correction. ****: p ≤ 0.0001, *: p ≤ 0.05.

The online version of this article includes the following source data for figure 3:

**Source data 1.** Related to *Figure 3*.

in the clustering analysis (see Methods; *Alexander-Bloch et al., 2013*; *Whitwell et al., 2009*). Two brain aging patterns were identified, where 18,929 (51.1%) participants with the first brain aging pattern (pattern 1) had higher total GMV at baseline and a slower rate of GMV decrease over time, and the remaining participants with the second pattern (pattern 2) had lower total GMV at baseline and a faster rate of GMV decrease (*Figure 2a*). Comparing the region-specific rate of GMV decrease, pattern 2 showed a more rapid GMV decrease in medial occipital (lingual gyrus, cuneus, and perical-carine cortex) and medial temporal (entorhinal cortex, parahippocampal gyrus) regions (*Figure 2b and c* and *Appendix 1—figure 2*), which had the largest loadings in the second and third principal components (*Appendix 1—table 4*). These two patterns can be clearly stratified by both linear and non-linear dimensionality reduction methods, indicating distinct structural differences in brain aging between patterns (*Appendix 1—figure 3*). Sample characteristics of these 37,013 UK Biobank participants stratified by brain aging patterns are summarized in *Appendix 1—table 5*. Overall, participants with different brain aging patterns had similar distributions with regard to age, sex, ethnicity, smoking status, Townsend deprivation index (TDI), body mass index (BMI) and years of schooling.

## Brain aging patterns were significantly associated with biological aging

To explore the relationships between structural brain aging and biological aging, we investigated the distribution of aging biomarkers, such as telomere length and PhenoAge (*Levine et al., 2018*), across brain aging patterns identified above (*Figure 3* and *Appendix 1—table 6*). Compared to pattern 1, participants in pattern 2 with more rapid GMV decrease had shorter leucocyte telomere length (p=0.009, Cohen's D=–0.028) and this association remained consistent after adjusting for sex, age, ethnic, BMI, smoking status and alcohol intake frequency (*Demanelis et al., 2020*). Next, we examined PhenoAge, which was developed as an aging biomarker incorporating composite clinical and biochemical data (*Levine et al., 2018*), and observed higher PhenoAge among participants with

brain aging pattern 2 compared to pattern 1 (p=0.019, Cohen's D=0.027). Again, the association remained significant after adjusting for sex, age, ethnic, BMI, smoking status, alcohol intake frequency and education years (p=3.05×10$^{-15}$, Cohen's D=0.092). Group differences in terms of each individual component of PhenoAge (including albumin, creatinine, glucose, c-reactive protein, lymphocytes percentage, mean corpuscular volume, erythocyte distribution width, alkaline phosphatase and leukocyte count) were also investigated and results were consistent with PhenoAge (*Appendix 1—figure 4*).

### Accelerated brain aging was associated with cognitive decline and increased genetic susceptibilities to attention-deficit/hyperactivity disorder and delayed brain development

Next, we conducted comprehensive comparisons of cognitive functions between participants with different brain aging patterns. In general, those with brain aging pattern 2 (lower baseline total GMV and more rapid GMV decrease) exhibited worse cognitive performances compared to pattern 1. Specifically, brain aging pattern 2 showed lower numbers of correct pairs matching (p=0.006, Cohen's D=−0.029), worse prospective memory (OR = 0.943, 95% CI [0.891, 0.999]), lower fluid intelligence (p<1.00×10$^{-20}$, Cohen's D=−0.102), and worse numeric memory (p=5.97×10$^{-11}$, Cohen's D=−0.082). No statistically significant differences were observed in terms of the reaction time (p=0.99) and

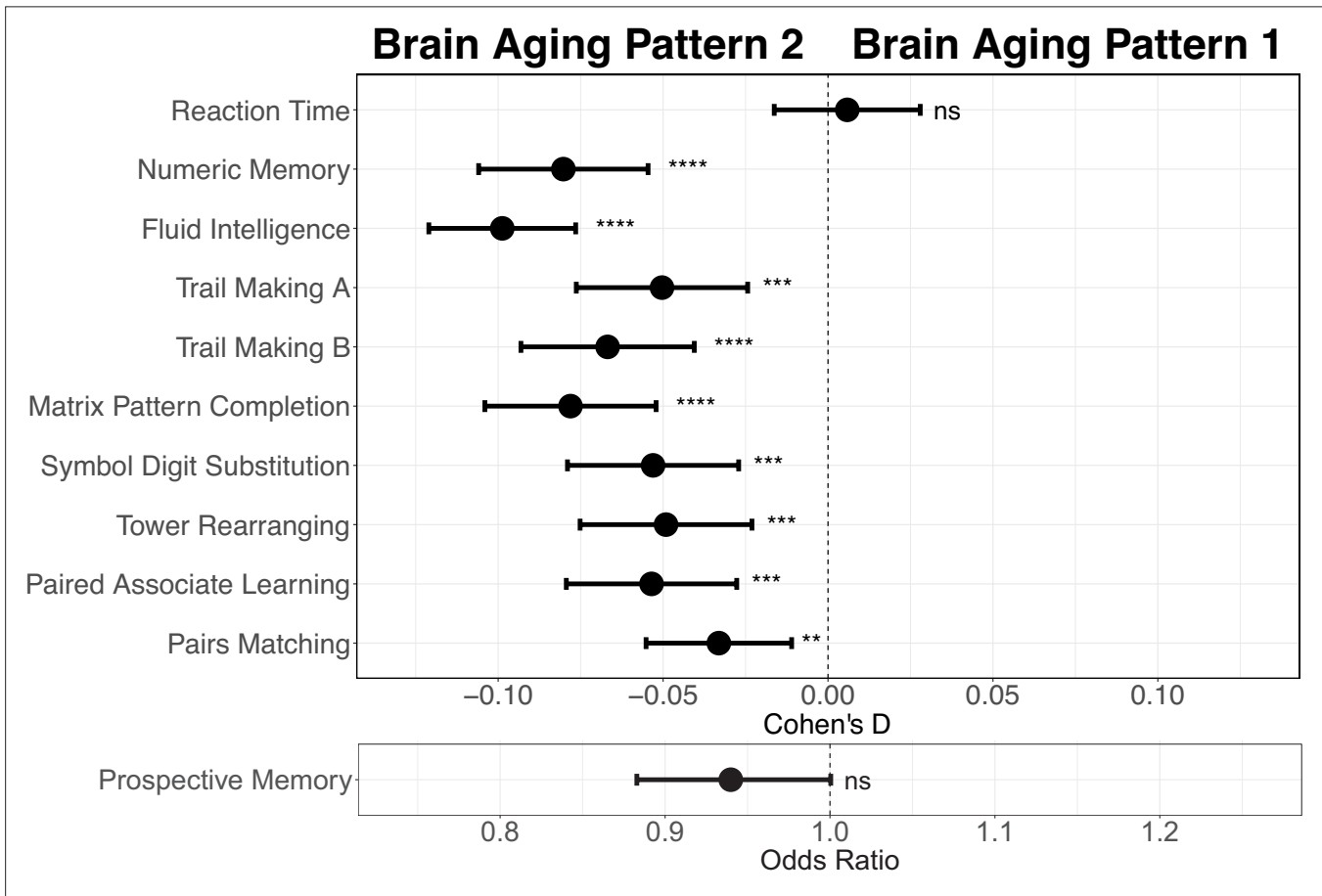

**Figure 4.** Effect size (Cohen's D or odds ratio) for comparing the cognitive functions between participants with brain aging patterns 1 and 2. Results were adjusted such that negative Cohen's D and Odds Ratio less than 1 indicate worse cognitive performances in brain aging pattern 2 compared to pattern 1. Width of the lines extending from the center point represent 95% confidence interval. Two-sided p values were obtained using both unadjusted and adjusted (for sex, age, and TDI, education and income) multivariate regression models. Stars indicate statistical significance after FDR correction for 11 comparisons. ****: p ≤ 0.0001, ***: p ≤ 0.001, **: p ≤ 0.01, ns: p>0.05.

The online version of this article includes the following source data for figure 4:

**Source data 1.** Related to *Figure 4*.

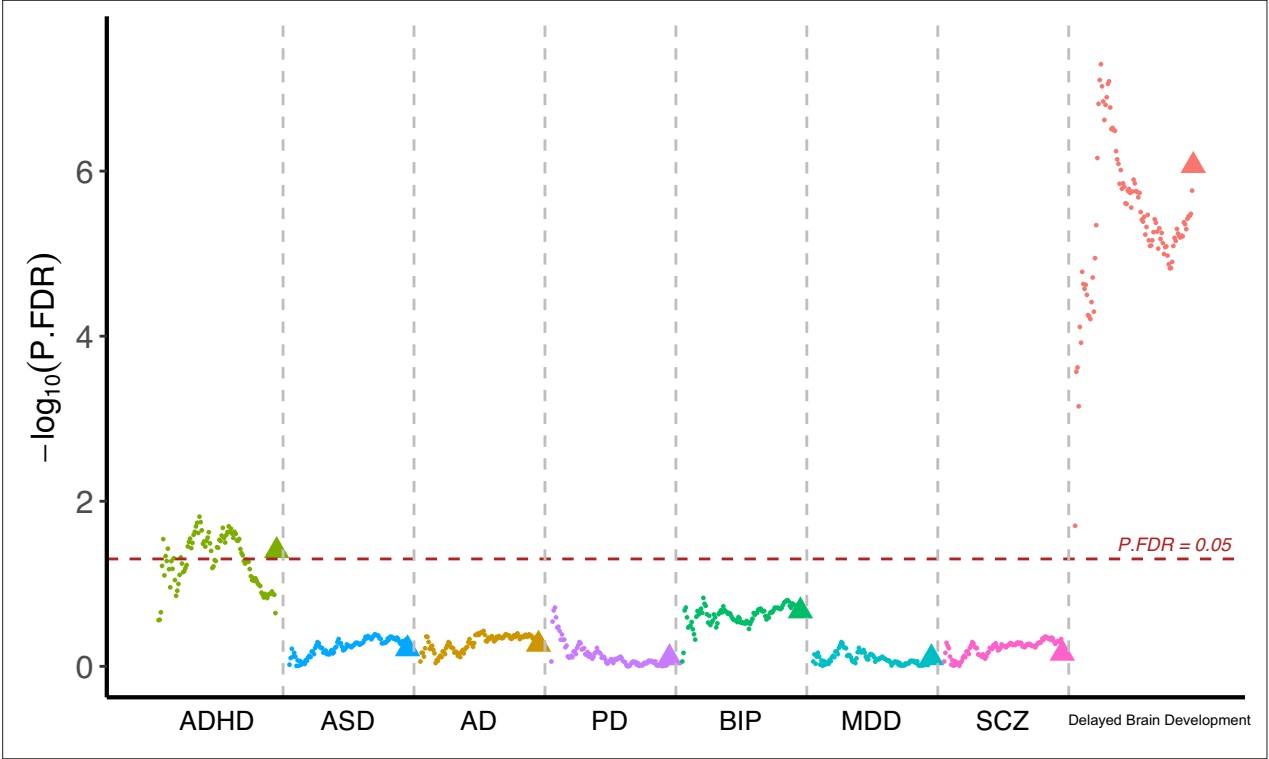

**Figure 5.** Participants with accelerated brain aging (brain aging pattern 2) had significantly increased genetic liability to ADHD and delayed brain development. Polygenic risk score (PRS) for ADHD, ASD, AD, PD, BIP, MDD, SCZ and delayed brain development (unpublished GWAS) were calculated at different p-value thresholds from 0.005 to 0.5 at an interval of 0.005. Vertical axis represents negative logarithm of P values comparing PRS in brain aging pattern 2 relative to pattern 1. Red horizontal dashed line indicates FDR corrected p value of 0.05. Colors represent traits and dots within the same color represent different p value thresholds. The trigonometric symbol indicates the average PRS across all p-value thresholds for the same trait.

The online version of this article includes the following source data for figure 5:

**Source data 1.** Related to *Figure 5*.

prospective memory (p=0.052) between these two brain aging patterns after FDR correction. Results were consistent when using models adjusted for sex, age, and socioeconomic status (TDI, education and income; *Foster et al., 2018*; *Townsend et al., 2023*; *Figure 4*). Full results demonstrating the associations between brain aging patterns and cognitive functions are presented in *Appendix 1—table 7*.

Having observed cognitive decline among participants with accelerated brain aging pattern, we next investigated whether brain aging patterns were associated with genetic vulnerability to major neuropsychiatric disorders. Since current GWAS are under-powered for attention-deficit/hyperactivity disorder (ADHD) and autism spectrum disorders (ASD) and the difficulty in identifying genetic variants was likely due to their polygenic nature, we calculated the corresponding polygenic risk scores (PRS) using multiple p value thresholds. This approach enabled robust investigation of the association between genetic susceptibility of neuropsychiatric disorders and brain imaging phenotypes. PRS for major neuro-developmental disorders including attention-deficit/hyperactivity disorder (ADHD) and autism spectrum disorders (ASD), neurodegenerative diseases including Alzheimer's disease (AD) and Parkinson's disease (PD), neuropsychiatric disorders including bipolar disorder (BIP), major depressive disorder (MDD), and schizophrenia (SCZ), and delayed structural brain development (GWAS from an unpublished longitudinal neuroimaging study) (*Shi et al., 2023*) were calculated for each participant using multiple p value thresholds (from 0.005 to 0.5 at intervals of 0.005) and results were then averaged over all thresholds (*Figure 5*). The primary GWAS datasets used for calculating the PRS were listed in *Appendix 1—table 8*. Overall, we observed increased genetic susceptibility to ADHD (p=0.040) and delayed brain development (p=$1.48 \times 10^{-6}$) among participants with brain aging pattern 2 after FDR correction, while no statistically significant differences were observed for ASD, AD, PD,

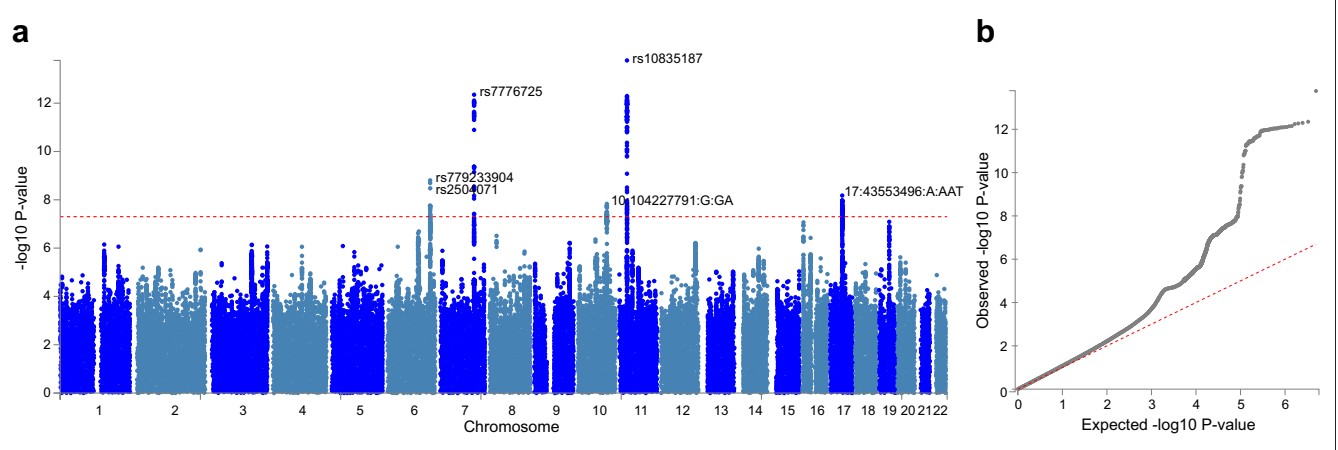

**Figure 6.** Genome-wide association study (GWAS) identified 6 independent SNPs associated with accelerated brain aging. Total GMV at 60 years old was estimated for each participant using mixed effect models allowing for individualized baseline GMV and GMV change rate, and was used as the phenotype in the GWAS. (**a**) At genome-wide significance level (p=5×10⁻⁸, red dashed line), rs10835187 and rs7776725 loci were identified to be associated with accelerated brain aging. (**b**) Quantile–quantile plot showed that the most significant p values deviate from the null, suggesting that results are not unduly inflated.

BIP, MDD, and SCZ (*Figure 5*). Details regarding the genetic liability to other common diseases and phenotypes using enhanced PRS from UK Biobank are displayed in *Appendix 1tables 9 and 10*.

## Genome Wide Association Studies (GWAS) identified significant genetic loci associated with accelerated brain aging

Having observed significant associations between brain aging patterns and cognitive performances / genetic liabilities to major neurodevelopmental disorders, we further investigated if there exist genetic variants contributing to individualized brain aging phenotype. We conducted genome-wide association studies (GWAS) using estimated total GMV at 60 years old as the phenotype. This phenotype was derived by adding individual specific deviations to the population averaged total GMV, thus providing additional information compared to studies using only cross-sectional neuroimaging phenotypes.

Six independent single nucleotide polymorphisms (SNPs) were identified at genome-wide significance level (p<5×10⁻⁸) (*Figure 6*) and were subsequently mapped to genes using NCBI, Ensembl and UCSC Genome Browser database (*Appendix 1—table 11*). Among them, two SNPs (rs10835187 and rs779233904) were also found to be associated with multiple brain imaging phenotypes in previous studies (*Smith et al., 2021*), such as regional and tissue volume, cortical area and white matter tract measurements. Compared to the GWAS using global gray matter volume as the phenotype, our GWAS revealed additional signal in chromosome 7 (rs7776725), which was mapped to the intron of FAM3C and encodes a secreted protein involved in pancreatic cancer (*Grønborg et al., 2006*) and Alzheimer's disease (*Liu et al., 2016*). This signal was further validated to be associated with specific brain aging mode by another study using a data-driven decomposition approach (*Smith et al., 2020*). In addition, another significant loci (rs10835187, p=1.11×10⁻¹³) is an intergenic variant between gene LGR4-AS1 and LIN7C, and was reported to be associated with bone density and brain volume measurement (*Smith et al., 2021*; *Estrada et al., 2012*). *LIN7C* encodes the Lin-7C protein, which is involved in the localization and stabilization of ion channels in polarized cells, such as neurons and epithelial cells (*Bohl et al., 2007*; *Kaech et al., 1998*). Previous study has revealed the association of both allelic and haplotypic variations in the *LIN7C* gene with ADHD (*Lanktree et al., 2008*).

## Mirroring patterns between brain aging and brain development

Having observed significant associations between brain aging and genetic susceptibility to neurodevelopmental disorders, we are now interested in examining the mirroring patterns between brain aging and brain development in the whole population, and whether these mirroring patterns were more pronounced in those with accelerated brain aging. Adolescents in the IMAGEN cohort showed

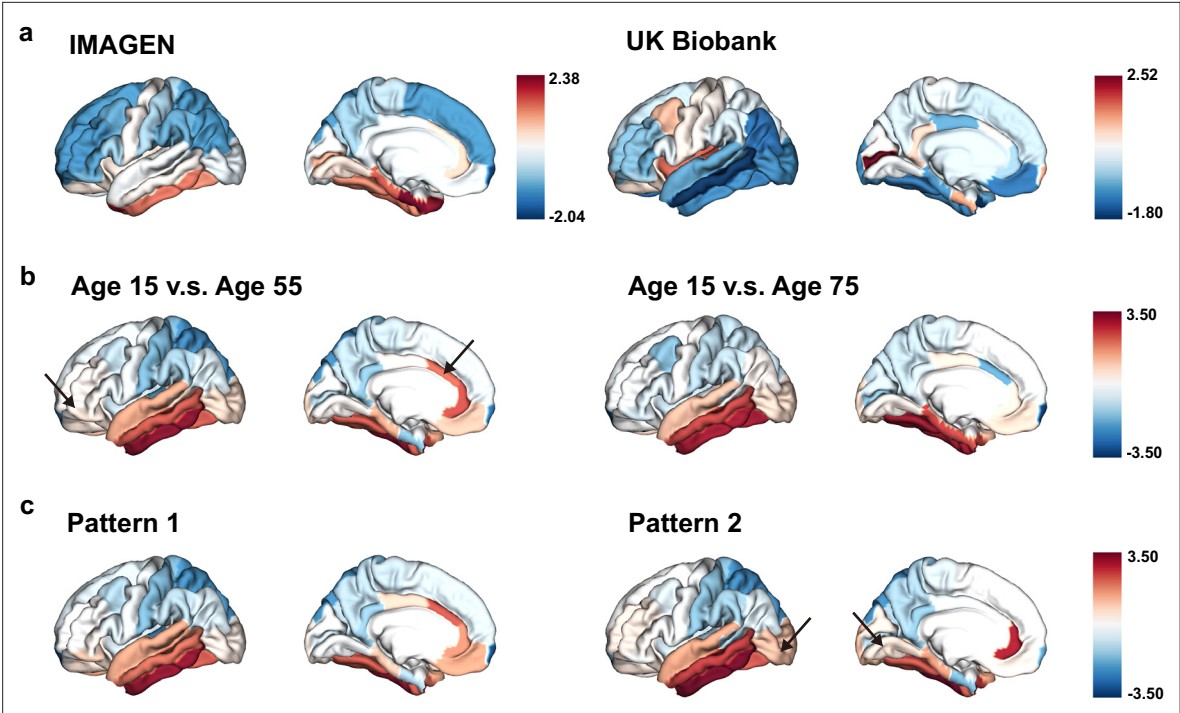

**Figure 7.** The 'last in, first out' mirroring patterns between brain development and brain aging. (**a**) The annual percentage volume change (APC) was calculated for each ROI and standardized across the whole brain in adolescents (IMAGEN, left) and mid-to-late aged adults (UK Biobank, right), respectively. For adolescents, ROIs of in red indicate delayed structural brain development, while for mid-to-late aged adults, ROIs in blue indicate accelerated structural brain aging. (**b**) Estimated APC in brain development versus early aging (55 years old, left), and versus late aging (75 years old, right). ROIs in red indicate faster GMV decrease during brain aging and slower GMV decrease during brain development, that is stronger mirroring effects between brain development and brain aging. (**c**) Mirroring patterns between brain development and brain aging were more manifested in participants with accelerated aging (brain aging pattern 2). The arrows point to ROIs with more pronounced mirroring patterns in each subfigure.

The online version of this article includes the following source data for figure 7:

**Source data 1.** Related to *Figure 7*.

more rapid GMV decrease in the frontal and parietal lobes, especially the frontal pole, superior frontal gyrus, rostral middle frontal gyrus, inferior parietal lobule and superior parietal lobule, while those in their mid-to-late adulthood showed more accelerated GMV decrease in the temporal lobe, including medial orbitofrontal cortex, inferior parietal lobule and lateral occipital sulcus (*Figure 7a*). The mirroring patterns (with slower GMV decrease during brain development and more rapid GMV decrease during brain aging) were particularly prominent in inferior temporal gyrus, caudal anterior cingulate cortex, fusiform cortex, middle temporal gyrus and rostral anterior cingulate cortex (*Figure 7b*). The regional mirroring patterns became weaker when we focus on late brain aging at age 75 years old, especially in the frontal lobe and cingulate cortex. Further, mirroring patterns were represented more prominently in participants with brain aging pattern 2, where stronger mirroring between brain aging and brain development was observed in frontotemporal area, including lateral occipital sulcus and lingual gyrus (*Figure 7c*).

## Gene expression profiles were associated with delayed brain development and accelerated brain aging

The Allen Human Brain Atlas (AHBA) transcriptomic dataset (http://human.brain-map.org) were used to obtain the spatial correlation between gene expression profiles across cortex and structural brain development/aging via partial least square (PLS) regression. The first PLS component explained 24.7% and 53.6% of the GMV change during brain development (estimated at age 15y, $r_{spearman}$ = 0.51, $P_{permutation}$ = 0.03) and brain aging (estimated at age 55y, $r_{spearman}$ = 0.49, $P_{permutation}$ = 1.5×10$^{-4}$), respectively. Seventeen of the 45 genes mapped to GWAS significant SNP were found in AHBA, with *LGR4* ($r_{spearman}$ = 0.56, $P_{permutation}$ <0.001) significantly associated with delayed brain development and *ESR1* ($r_{spearman}$

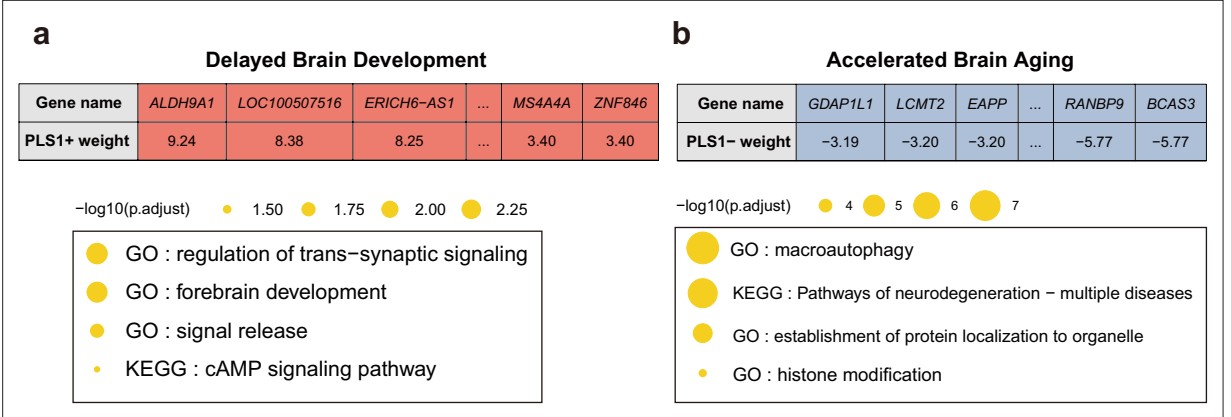

**Figure 8.** Functional enrichment of gene transcripts significantly associated with delayed brain development and accelerated brain aging. (**a**) 990 genes were spatially correlated with the first PLS component of delayed structural brain development, and were enriched for trans-synaptic signal regulation, forebrain development, signal release and cAMP signaling pathway. (**b**) 2293 genes were spatially correlated the first PLS component of accelerated structural brain aging, and were enriched for macroautophagy, pathways of neurodegeneration, establishment of protein localization to organelle and histone modification. Size of the circle represents number of genes in each term and P values were corrected using FDR for multiple comparisons.

The online version of this article includes the following source data for figure 8:

**Source data 1.** Related to *Figure 8*.

**Source data 2.** Related to *Figure 8*.

= 0.53, $P_{permutation}$ <0.001) and *FAM3C* ($r_{spearman}$ = −0.37, $P_{permutation}$ = 0.004) significantly associated with accelerated brain aging. *BDNF-AS* was positively associated with both delayed brain development and accelerated brain aging after spatial permutation test (*Appendix 1—table 12 and 13*).

Next, we screened the genes based on their contributions and effect directions to the first PLS components in brain development and brain aging. 990 and 2293 genes were identified to be positively associated with brain development and negatively associated with brain aging at FDR corrected p value of 0.005, respectively, representing gene expressions associated with delayed brain development and accelerated brain aging. These genes were then tested for enrichment of GO biological processes and KEGG pathways. Genes associated with delayed brain development showed significant enrichment in 'regulation of trans-synaptic signaling', 'forebrain development', 'signal release' and 'cAMP signaling pathway' (*Figure 8a*), and genes associated with accelerated brain aging showed significant enrichment in 'macroautophagy', 'establishment of protein localization to organelle', 'histone modification', and 'pathways of neurodegeneration – multiple diseases' (*Figure 8b*). Full results of the gene set enrichment analysis were provided in *Appendix 1—figure 5*. In summary, the analyses from using the databases of GO biological processes and KEGG Pathways indicate synaptic transmission as an important process in the common mechanisms of brain development and aging, and cellular processes (autophagy), as well as the progression of neurodegenerative diseases, are important processes in the mechanisms of brain aging.

## Discussion

In this study, we adopted a data-driven approach and revealed two distinct brain aging patterns using large-scale longitudinal neuroimaging data in mid-to-late adulthood. Compared to brain aging pattern 1, brain aging pattern 2 were characterized by a faster rate of GMV decrease, accelerated biological aging, cognitive decline, and genetic susceptibility to neurodevelopmental disorders. By integrating longitudinal neuroimaging data from adult and adolescent cohorts, we demonstrated the 'last in, first out' mirroring patterns between structural brain aging and brain development, and showed that the mirroring pattern was manifested in the temporal lobe and among participants with accelerated brain aging. Further, genome-wide association studies identified significant genetic loci contributing to accelerated brain aging, while spatial correlation between whole-brain transcriptomic profiles and structural brain aging / development revealed important gene sets associated with both accelerated brain aging and delayed brain development.

Brain aging is closely related to the onset and progression of neurodegenerative and neuropsychiatric disorders. Both neurodegenerative and neuropsychiatric disorders demonstrate strong inter-individual heterogeneity, which prevents the comprehensive understanding of their neuropathology and neurogenetic basis. Therefore, multidimensional investigation into disease subtyping and population clustering of structural brain aging are crucial in elucidating the sources of heterogeneity and neurophysiological basis related to the disease spectrum (*Habes et al., 2020*). In the last decades, major developments in the subtyping of Alzheimer's disease, dementia and Parkinson's disease, have provided new perspectives regarding their clinical diagnosis, treatment, disease progression and prognostics (*Habes et al., 2020*; *Berg et al., 2021*; *Ferreira et al., 2020*). While previous studies of brain aging mostly focused on the cross-sectional differences between cases and healthy controls, we here delineated the structural brain aging patterns among healthy participants using a novel data-driven approach that captured both cross-sectional and longitudinal trajectories of the whole-brain gray matter volume (*Feczko et al., 2019*; *Poulakis et al., 2022*). The two brain aging patterns identified using the above approach showed large differences in the rate of change in medial occipitotemporal gyrus, which is involved in vision, word processing, and scene recognition (*Bogousslavsky et al., 1987*; *Epstein et al., 1999*; *Mechelli et al., 2000*). Significant reduction of the gray matter volume and abnormal changes of the functional connectivity in this region were found in subjects with mild cognitive impairment (MCI) and AD, respectively (*Chételat et al., 2005*; *Yao et al., 2010*). Previous research on brainAGE (*Elliott et al., 2021*; *Christman et al., 2020*) (the difference between chronological age and the age predicted by the machine learning model of brain imaging data) showed that as a biomarker of accelerated brain aging, people with older brainAGE have accelerated biological aging and early signs of cognitive decline, which is consistent with our discoveries in this study. Our results support the establishment of a network connecting brain aging patterns with biological aging profiles involving multi-organ systems throughout the body (*Tian et al., 2023*). Since structural brain patterns might manifest and diverge decades before cognitive decline (*Aljondi et al., 2019*), subtyping of brain aging patterns could aid in the early prediction of cognitive decline and severe neurodegenerative and neuropsychiatric disorders.

Mirroring pattern between brain development and brain aging has long been hypothesized by postulating that phylogenetically newer and ontogenetically less precocious brain structures degenerate relatively early (*Douaud et al., 2014*). Early studies have reported a positive correlation between age-related differences of cortical volumes and precedence of myelination of intracortical fibers (*Raz, 2000*). Large differences in the patterns of change between adolescent late development and aging in the medial temporal cortex were previously found in studies of brain development and aging patterns (*Tamnes et al., 2013*). Here, we compared the annual volume change of the whole-brain gray matter during brain development and early / late stages of brain aging, and found that mirroring patterns are predominantly localized to the lateral / medial temporal cortex and the cingulate cortex. These cortical regions characterized by 'last in, first out' mirroring patterns showed increased vulnerability to the several neuropsychiatric disorders. For example, regional deficits in the superior temporal gyrus and medial temporal lobe were observed in schizophrenia (*Honea et al., 2005*), along with morphological abnormalities in the medial occipitotemporal gyrus (*Schultz et al., 2010*). Children diagnosed with ADHD had lower brain surface area in the frontal, cingulate, and temporal regions (*Hoogman et al., 2019*). *Douaud et al., 2014* revealed a population transmodal network with lifespan trajectories characterized by the mirroring pattern of development and aging. We investigated the genetic susceptibility to individual-level mirroring patterns based on the lasting impact of neurodevelopmental genetic factors on brain (*Fjell et al., 2015*), demonstrating that those with more rapidly brain aging patterns have a higher risk of delayed development.

Identifying genes contributing to structural brain aging remains a critical step in understanding the molecular changes and biological mechanisms that govern age-related cognitive decline. Several genetic loci have been reported to be associated with brain aging modes and neurocognitive decline, many of which demonstrated global overlap with neuropsychiatric disorders and their related risk factors (*Smith et al., 2020*; *Glahn et al., 2013*; *Brouwer et al., 2022*). Here, we focused on the individual brain aging phenotype by estimating individual deviation from the population averaged total GMV and conducted genome-wide association analysis with this phenotype. Our approach identified six risk SNPs associated with accelerated brain aging, most of which could be further validated by previous studies using population averaged brain aging phenotypes. However, our approach revealed

additional genetic signals and demonstrated genetic architecture underlying brain aging patterns overlap with bone density (*Estrada et al., 2012*; *Zheng et al., 2015*). In addition, molecular profiling of the aging brain has been thoroughly investigated among patients with neurogenerative diseases, but rarely conducted to shed light on the mirroring patterns among healthy participants. Analysis of the spatial correlation between gene expression profiles and structural brain development / aging further identified genes contributing to delayed brain development and accelerated brain aging. Specifically, expression of gene *BDNF-AS* was significantly associated with both processes. *BDNF-AS* is an antisense RNA gene and plays a role in the pathoetiology of non-neoplastic conditions mainly through the mediation of *BDNF* (*Ghafouri-Fard et al., 2021*). LGR4 (associated with delayed brain development) and FAM3C (associated with accelerated brain aging) identified in the spatial genetic association analysis also validated our findings in the GWAS.

There are several limitations in the current study that need to be addressed in future research. Firstly, the UK Biobank cohort, which we leveraged to identify population clustering of brain aging patterns, had a limited number of repeated structural MRI scans. Therefore, it remains challenging to obtain robust estimation of the longitudinal whole-brain GMV trajectory at the individual level. As a robustness check, we have calculated both intra-class correlation and variance of both random intercept and age slope to ensure appropriateness of the mixed effect models. Secondly, although aging is driven by numerous hallmarks, we have only investigated the association between brain aging patterns and biological aging in terms of telomere length and blood biochemical markers due to limitations of data access. Other dimensions of aging hallmarks and their relationship with structural brain aging need to be investigated in the future. Thirdly, our genomic analyses were restricted to 'white British' participants of European ancestry. The diversity of genomic analyses will continue to improve as the sample sizes of GWAS of non-European ancestry increase. Further, although the gene expression maps from Allen Human Brain Atlas enabled us to gain insights into the spatial coupling between gene expression profiles and mirroring patterns of the brain, the strong inter-individual variation of whole-brain gene expression levels and large temporal span of the human brain samples may lead to the inaccurate correspondence in the observed associations. Finally, we focused on structural MRIs in deriving brain aging patterns in this analysis, future investigations could consider other brain imaging modalities from a multi-dimensional perspective. Nevertheless, our study represents a novel attempt for population clustering of structural brain aging and validated the mirroring pattern hypothesis by leveraging large-scale adolescent and adult cohorts.

## Methods
### Participants
T1-weighted brain MRI images were obtained from 37,013 individuals aged 44–82 years old from UK Biobank (36,914 participants at baseline visit in 2014+, 4007 participants at the first follow-up visit in 2019+). All participants from UK Biobank provided written informed consent, and ethical approval was granted by the North West Multi-Center Ethics committee (https://www.ukbiobank.ac.uk/learn-more-about-uk-biobank/about-us/ethics) with research ethics committee (REC) approval number 16/NW/0274. Participants were excluded if they were diagnosed with severe psychiatric disorders or neurological diseases using ICD-10 primary and secondary diagnostic codes or from self-reported medical conditions at UK Biobank assessment center (see *Appendix 1—tables 1 and 2*). Data were obtained under application number 19542. A total of 1529 adolescents with structural MRI images were drawn from the longitudinal project IMAGEN (1463 at age 14, 1377 at age 19, and 1148 at age 23), of which the average number of MRI scans was 2.61 per adolescent. The IMAGEN study was approved by local ethics research committees of King's College London, University of Nottingham. Trinity College Dublin, University of Heidelberg, Technische Universität Dresden, Commissariat à l'Énergie Atomique et aux Énergies Alternatives, and University Medical Center at the University of Hamburg in compliance with the Declaration of Helsinki (*Association, 2013*). Informed consent was given by all participants and a parent/guardian of each participant.

### MRI acquisition
Quality-controlled T1-weighted neuroimaging data from UK Biobank and IMAGEN were processed using FreeSurfer v6.0. Detailed imaging processing pipeline can be found online for UK Biobank

([https://biobank.ctsu.ox.ac.uk/crystal/crystal/docs/brain_mri.pdf](https://biobank.ctsu.ox.ac.uk/crystal/crystal/docs/brain_mri.pdf)) and IMAGEN ([https://github.com/imagen2/imagen_mri](https://github.com/imagen2/imagen_mri); *Schumann et al., 2010*; *Imagen, 2020*). Briefly, cortical gray matter volume (GMV) from 33 regions in each hemisphere were generated using Desikan–Killiany Atlas (*Desikan et al., 2006*), and total gray matter volume (TGMV), intracranial volume (ICV) and subcortical volume were derived from ASEG atlas (*Fischl et al., 2002* See *Appendix 1—table 3*). Regional volume was averaged across left and right hemispheres. To avoid deficient segmentation or parcellation, participants with TGMV, ICV or regional GMV beyond 4 standard deviations from the sample mean were considered as outliers and removed from the following analyses.

## Identification of longitudinal brain aging patterns

Whole-brain GMV trajectory was estimated for each participant in 40 brain regions of interest (ROIs) (33 cortical regions and 7 subcortical regions), using mixed effect regression model with fixed linear and quadratic age effects, random intercept and random age slope. Covariates include sex, assessment center, handedness, ethnic, and ICV. Models with random intercept and with both random intercept and random age slope were compared using AIC, BIC and evaluation of intra-class correlation (ICC). Results suggested that random age slope model should be chosen for almost all ROIs (*Appendix 1—table 14*). Deviation of regional GMV from the population average was calculated for each participant at age 60 years and dimensionality reduction was conducted via principal component analysis (PCA). The first 15 principal components explaining approximately 70% of the total variations of regional GMV deviation were used in multivariate k-means clustering. Optimal number of clusters was chosen using both elbow diagram and contour coefficient (*Appendix 1—figure 6*). Rates of volumetric change for total gray matter and each ROI were estimated using generalized additive mixed effect models (GAMM) with fixed cubic splines of age, random intercept and random age slope, which incorporates both cross-sectional between-subject variation and longitudinal within-subject variation from 40,921 observations and 37,013 participants. Covariates include sex, assessment center, handedness, ethnic, and ICV. We also applied PCA and locally linear embedding (LLE; *Roweis and Saul, 2000*) to the adjusted GMV ROIs in order to map the high-dimensional imaging-derived phenotypes to a low-dimensional space for stratification visualisation. The GMV of 40 ROIs at baseline were linearly adjusted for sex, assessment center, handedness, ethnic, ICV, and second-degree polynomial in age to be consistent with the whole-brain GMV trajectory model.

## Association between brain aging patterns and biological aging, cognitive decline and genetic susceptibilities of neuropsychiatric disorders

Individuals with Z-standardized leucocyte telomere length (*Codd et al., 2021*) and blood biochemistry (which were used to calculate PhenoAge (*Levine et al., 2018*) that characterizes biological aging) outside 4 standard deviations from the sample mean were excluded for better quality control. A total of 11 cognitive tests performed on the touchscreen questionnaire were included in the analysis. More information about the cognitive tests is provided in Supplementary Information. Comparisons of biological aging (leucocyte telomere length, PhenoAge) and cognitive function were conducted among participants with different brain aging patterns using both unadjusted and adjusted multivariate regression models with Bonferroni / FDR correction. Polygenic Risk Scores (PRS) were calculated for autism spectrum disorder (ASD), attention deficit hyperactivity disorder (ADHD), Alzheimer's disease (AD), Parkinson's Disease (PD), bipolar disorder (BIP), major depressive disorder (MDD), schizophrenia (SCZ), and delayed brain development using GWAS summary statistics (*Shi et al., 2023*) at multiple p value thresholds (from 0.005 to 0.5 at intervals of 0.005, and 1), with higher p value thresholds incorporating larger number of independent SNPs. After quality control of genotype and imaging data, PRSs were generated for 25,861 participants on UK Biobank genotyping data. SNPs were pruned and clumped with a cutoff $r^2 \geq 0.1$ within a 250 kb window. All calculations were conducted using PRSice v2.3.5 (*Choi and O'Reilly, 2019*). Enhanced PRS from UK Biobank Genomics for multiple diseases were also tested. Detailed instructions for calculating enhanced PRS in UK Biobank can be found in research of *Thompson et al., 2022* Comparisons of neuropsychiatric disorders were conducted among participants with different brain aging patterns using t test with FDR correction. All statistical tests were two-sided.

## Genome wide association Study to identify SNPs associated with brain aging patterns

We performed Genome-wide association studies (GWAS) on individual deviations of total GMV relative to the population average at 60 years using PLINK 2.0 (*Chang et al., 2015*). Variants with missing call rates exceeding 5%, minor allele frequency below 0.5% and imputation INFO score less than 0.8 were filtered out after the genotyping quality control for UK Biobank Imputation V3 dataset. Among the 337,138 unrelated 'white British' participants of European ancestry included in our study, 25,861 with recent UK ancestry and accepted genotyping and imaging quality control were included in the GWAS. The analyses were further adjusted for age, age2, sex, assessment center, handedness, ethnic, ICV, and the first 10 genetic principal components. Genome-wide significant SNPs ($p < 5 \times 10^{-8}$) obtained from the GWAS were clumped by linkage disequilibrium (LD) ($r^2 < 0.1$ within a 250 kb window) using UKB release2b White British as the reference panel. We subsequently performed gene-based annotation in FUMA (*Watanabe et al., 2017*) using genome-wide significant SNPs and SNPs in close LD ($r^2 \geq 0.1$) using Annotate Variation (ANNOVAR) on Ensemble v102 genes (*Wang et al., 2010*).

## Mirroring patterns between brain aging and brain development

To validate the 'last in, first out' mirroring hypothesis, we evaluated the structural association between brain development and brain aging. Longitudinal neuroimaging data from 1529 adolescents in the IMGAEN cohort and 3908 mid-to-late adulthood in the UK Biobank cohort were analyzed. Annual percentage volume change (APC) for each ROI was calculated among individuals with at least two structural MRI scans by subtracting the baseline GMV from follow-up GMV and dividing by the number of years between baseline and follow-up visits. Region-specific APC was regressed on age using smoothing spline with cross validated degree of freedom. Estimated APC for each ROI was obtained at age 15y for adolescents and at age 55y (early aging) and 75y (late aging) for participants in UK Biobank. Region-specific APC during adolescence (or mid-to-late adulthood) was then standardized across all cortical regions to create the brain development (or aging) map. Finally, the brain development map and brain aging map were compared to assess the mirroring pattern for each ROI in the overall population and across different aging subgroups.

## Gene expression analysis

The Allen Human Brain Atlas (AHBA) dataset (http://human.brain-map.org), which comprises gene expression measurements in six postmortem adults (age 24–57y) across 83 parcellated brain regions (*Hawrylycz et al., 2012*; *Markello et al., 2021*), were used to identify gene expressions significantly associated with structural brain development and aging. The expression profiles of 15,633 genes were averaged across donors to form a $83 \times 15,633$ transcriptional matrix and partial least squares (PLS) regression was adopted for analyzing the association between regional change rate of gray matter volume and gene expression profiles. Specifically, estimated regional APC at 15 (obtained from IMAGEN cohort) and 55 years old (obtained from UK Biobank) were regressed on the high-dimensional gene expression profiles upon regularization. Associations between the first PLS component and estimated APC during brain development and brain aging were tested by spatial permutation analysis (10,000 times; *Váša et al., 2018*). Additionally, gene expression profiles of genes mapped to GWAS significant SNP were extracted from AHBA. The association between gene expression profiles of mapped genes and estimated APC during brain development and aging was also tested by spatial permutation analysis. Statistical significance of each gene's contribution to the first PLS component was tested with standard error calculated using bootstrap (*Li et al., 2021*; *Morgan et al., 2019*; *Romero-Garcia et al., 2020*), and genes significantly associated with delayed brain development and accelerated brain aging were selected. Enrichment of Kyoto Encyclopedia of Genes and Genomes (KEGG) pathways and gene ontology (GO) of biological processes for these selected genes were analyzed using R package clusterProfiler (*Yu et al., 2012*). All statistical significances were corrected for multiple testing using FDR.

## Code availability

R version 4.2.0 was used to perform statistical analyses. FreeSurfer version 6.0 was used to process neuroimaging data. lme4 1.1 in R version 4.2.0 was used to perform longitudinal data analyses. PRSice version 2.3.5 (https://choishingwan.github.io/PRSice/; *Choi and O'Reilly, 2019*) was used to calculate

the PRS. PLINK 2.0 (https://www.cog-genomics.org/plink/2.0/) and FUMA version 1.5.6 (https://fuma.ctglab.nl/) were used to perform genome-wide association analysis, and ANNOVAR was used to perform gene-based annotation. AHBA microarray expression data were processed using abagen toolbox version 0.1.3 (https://doi.org/10.5281/zenodo.5129257). The rotate_parcellation code used to perform a spatial permutation test of a parcellated cortical map: https://github.com/frantisekvasa/rotate_parcellation (*Váša, 2023*; *Váša et al., 2018*). Code for PLS analysis and bootstrapping to estimate PLS weights are available at https://github.com/KirstieJane/NSPN_WhitakerVertes_PNAS2016/tree/master/SCRIPTS (*Whitaker, 2016*; *Whitaker et al., 2016*). clusterProfiler 4.6 in R version 4.2.0 was used to analyze gene-set enrichment.

## Acknowledgements

This research used the UK Biobank Resource under application number 19542. We thank all participants and researchers from the UK Biobank. We thank the IMAGEN Consortium for providing the discover data. This work received support from the following sources: the National Nature Science Foundation of China (No.82304241 [to XL]), National Key R&D Program of China (No.2019YFA0709502 [to JF], No.2018YFC1312904 [to JF]), Shanghai Municipal Science and Technology Major Project (No.2018SHZDZX01 [to JF], ZJ Lab [to JF], and Shanghai Center for Brain Science and Brain-Inspired Technology [to JF]), the 111 Project (No.B18015 [to JF]), the European Union-funded FP6 Integrated Project IMAGEN (Reinforcement-related behaviour in normal brain function and psychopathology) (LSHM-CT- 2007–037286 [to GS]), the Horizon 2020 funded ERC Advanced Grant 'STRATIFY' (Brain network based stratification of reinforcement-related disorders) (695313 [to GS]), Human Brain Project (HBP SGA 2, 785907, and HBP SGA 3, 945539 [to GS]), the Medical Research Council Grant 'c-VEDA' (Consortium on Vulnerability to Externalizing Disorders and Addictions) (MR/N000390/1 [to GS]), the National Institute of Health (NIH) (R01DA049238 [to GS], A decentralized macro and micro gene-by-environment interaction analysis of substance use behavior and its brain biomarkers), the National Institute for Health Research (NIHR) Biomedical Research Centre at South London and Maudsley NHS Foundation Trust and King's College London, the Bundesministeriumfür Bildung und Forschung (BMBF grants 01GS08152; 01EV0711 [to GS]; Forschungsnetz AERIAL 01EE1406A, 01EE1406B; Forschungsnetz IMAC-Mind 01GL1745B [to GS]), the Deutsche Forschungsgemeinschaft (DFG grants SM 80/7–2, SFB 940, TRR 265, NE 1383/14–1 [to GS]), the Medical Research Foundation and Medical Research Council (grants MR/R00465X/1 and MR/S020306/1 [to SD]), the National Institutes of Health (NIH) funded ENIGMA (grants 5U54EB020403-05 and 1R56AG058854-01 [to SD]), NSFC grant 82150710554 [to GS] and European Union funded project 'environMENTAL', grant no: 101057429 [to GS]. Further support was provided by grants from: - the ANR (ANR-12-SAMA-0004, AAPG2019 - GeBra [to JLM]), the Eranet Neuron (AF12-NEUR0008-01 - WM2NA; and ANR-18-NEUR00002-01 - ADORe [to JLM]), the Fondation de France (00081242 [to J.-L.M.]), the Fondation pour la Recherche Médicale (DPA20140629802 [to JLM]), the Mission Interministérielle de Lutte-contre-les-Drogues-et-les-Conduites-Addictives (MILDECA [to JLM]), the Assistance-Publique-Hôpitaux-de-Paris and INSERM (interface grant [to MLPM]), Paris Sud University IDEX 2012 [to J.-LM], the Fondation de l'Avenir (grant AP-RM-17–013 [to MLPM]), the Fédération pour la Recherche sur le Cerveau [to GS]; the National Institutes of Health, Science Foundation Ireland (16/ERCD/3797 [to RW]) and by NIH Consortium grant U54 EB020403 [to SD], supported by a cross-NIH alliance that funds Big Data to Knowledge Centres of Excellence. The funders had no role in study design, data collection and analysis, decision to publish or preparation of the manuscript.

## Additional information

### Competing interests

Tobias Banaschewski: Dr Banaschewski served in an advisory or consultancy role for eye level, Infectopharm, Lundbeck, Medice, Neurim Pharmaceuticals, Oberberg GmbH, Roche, and Takeda. He received conference support or speaker's fee by Janssen, Medice and Takeda. He received royalties from Hogrefe, Kohlhammer, CIP Medien, Oxford University Press; the presentwork is unrelated to these relationships. Christian Büchel: Reviewing editor, *eLife*. Luise Poustka: Dr Poustka served in

an advisory or consultancy role for Roche and Viforpharm and received speaker's fee by Shire. She received royalties from Hogrefe, Kohlhammer and Schattauer. The present work is unrelated to the above grants and relationships. The other authors declare that no competing interests exist.

## Funding

| Funder | Grant reference number | Author |
|---|---|---|
| National Natural Science Foundation of China | No.82304241 | Xiaolei Lin |
| National Key Research and Development Program of China | No.2019YFA0709502 | Jianfeng Feng |
| National Key Research and Development Program of China | No.2018YFC1312904 | Jianfeng Feng |
| Shanghai Municipal Science and Technology Major Project | No.2018SHZDZX01 | Jianfeng Feng |
| ZJ Lab | | Jianfeng Feng |
| Shanghai Center for Brain Science and Brain-Inspired Technology | | Jianfeng Feng |
| 111 project | No.B18015 | Jianfeng Feng |
| European Union FP6 Integrated Project IMAGEN | LSHM-CT- 2007-037286 | Gunter Schumann |
| Horizon 2020 ERC Advanced Grant 'STRATIFY' | 695313 | Gunter Schumann |
| Human Brain Project | HBP SGA 2,785907 | Gunter Schumann |
| Human Brain Project | HBP SGA 3,945539 | Gunter Schumann |
| Medical Research Council Grant 'c-VEDA' | MR/N000390/1 | Gunter Schumann |
| National Institutes of Health | R01DA049238 | Gunter Schumann |
| National Institute for Health Research (NIHR) Biomedical Research Centre at South London and Maudsley NHS Foundation Trust and King's College London | | Gunter Schumann |
| Deutsche Forschungsgemeinschaft | SM 80/7-2 | Gunter Schumann |
| Medical Research Foundation and Medical Research Council | MR/R00465X/1 | Sylvane Desrivières |
| National Institutes of Health | ENIGMA 5U54EB020403-05 | Sylvane Desrivières |
| National Institutes of Health | U54 EB020403 | Sylvane Desrivières |
| ANR | ANR-12-SAMA-0004 AAPG2019 - GeBra | Jean-Luc Martinot |
| Eranet Neuron | AF12-NEUR0008-01 - WM2NA | Jean-Luc Martinot |
| Fondation de France | 00081242 | Jean-Luc Martinot |

| Funder | Grant reference number | Author |
| --- | --- | --- |
| Fondation pour la Recherche Médicale | DPA20140629802 | Jean-Luc Martinot |
| Mission Interministérielle de Lutte-contre-les-Drogues-et-les-Conduites-Addictives | MILDECA | Jean-Luc Martinot |
| Assistance-Publique-Hôpitaux-de-Paris and INSERM | interface grant | Marie-Laure Paillère Martinot |
| Paris Sud University IDEX 2012 | | Jean-Luc Martinot |
| Fondation de l'Avenir | AP-RM-17-013 | Marie-Laure Paillère Martinot |
| Fédération pour la Recherche sur le Cerveau | | Gunter Schumann |
| National Institutes of Health, Science Foundation Ireland | 16/ERCD/3797 | Robert Whelan |
| NSFC | 82150710554 | Gunter Schumann |
| environMENTAL | grant no: 101057429 | Gunter Schumann |
| Bundesministeriumfür Bildung und Forschung | 01GS08152 | Gunter Schumann |
| Bundesministeriumfür Bildung und Forschung | 01EV0711 | Gunter Schumann |
| Forschungsnetz AERIAL | 01EE1406A | Gunter Schumann |
| Forschungsnetz AERIAL | 01EE1406B | Gunter Schumann |
| Forschungsnetz IMAC-Mind | 01GL1745B | Gunter Schumann |
| Deutsche Forschungsgemeinschaft | SFB 940 | Gunter Schumann |
| Deutsche Forschungsgemeinschaft | TRR 265 | Gunter Schumann |
| Deutsche Forschungsgemeinschaft | NE 1383/14-1 | Gunter Schumann |
| Medical Research Foundation and Medical Research Council | MR/S020306/1 | Sylvane Desrivières |
| National Institutes of Health | ENIGMA 1R56AG058854-01 | Sylvane Desrivières |
| Eranet | ANR-18-NEUR00002-01 - ADORe | Jean-Luc Martinot |

The funders had no role in study design, data collection and interpretation, or the decision to submit the work for publication.

## Author contributions

Haojing Duan, Software, Formal analysis, Visualization, Methodology, Writing - original draft, Writing – review and editing; Runye Shi, Data curation, Visualization, Writing – review and editing; Jujiao Kang, Tobias Banaschewski, Arun LW Bokde, Christian Büchel, Herta Flor, Antoine Grigis, Hugh Garavan, Penny A Gowland, Andreas Heinz, Rüdiger Brühl, Eric Artiges, Frauke Nees, Dimitri Papadopoulos Orfanos, Luise Poustka, Sarah Hohmann, Nathalie Nathalie Holz, Juliane Fröhner, Michael N Smolka, Nilakshi Vaidya, Henrik Walter, Data curation, Writing – review and editing; Sylvane Desrivières, Jean-Luc Martinot, Marie-Laure Paillère Martinot, Robert Whelan, Gunter Schumann, Data curation,

Funding acquisition, Writing – review and editing; Xiaolei Lin, Conceptualization, Funding acquisition, Methodology, Writing - original draft, Writing – review and editing; Jianfeng Feng, Conceptualization, Funding acquisition, Writing – review and editing

## Author ORCIDs
Haojing Duan http://orcid.org/0009-0004-8659-9591
Christian Büchel https://orcid.org/0000-0003-1965-906X
Rüdiger Brühl https://orcid.org/0000-0003-0111-5996
Jean-Luc Martinot https://orcid.org/0000-0002-0136-0388
Dimitri Papadopoulos Orfanos https://orcid.org/0000-0002-1242-8990
Michael N Smolka https://orcid.org/0000-0001-5398-5569
Xiaolei Lin https://orcid.org/0000-0003-2463-1272
Jianfeng Feng https://orcid.org/0000-0001-5987-2258

## Ethics

All participants from UK Biobank provided written informed consent, and ethical approval was granted by the North West Multi-Center Ethics committee (https://www.ukbiobank.ac.uk/learn-more-about-uk-biobank/about-us/ethics) with research ethics committee (REC) approval number 16/NW/0274. The IMAGEN study was approved by local ethics research committees of King's College London, University of Nottingham. Trinity College Dublin, University of Heidelberg, Technische Universität Dresden, Commissariat à l'Énergie Atomique et aux Énergies Alternatives, and University Medical Center at the University of Hamburg in compliance with the Declaration of Helsinki (https://doi.org/10.1001/jama.2013.281053). Informed consent was given by all participants and a parent/guardian of each participant.

Reviewer #1 (Public Review): https://doi.org/10.7554/eLife.94970.3.sa1
Reviewer #2 (Public Review): https://doi.org/10.7554/eLife.94970.3.sa2
Author response https://doi.org/10.7554/eLife.94970.3.sa3

# Additional files

## Supplementary files
• MDAR checklist

## Data availability

The summary statistics of GWAS for individual deviations of total GMV is available at https://doi.org/10.5061/dryad.jh9w0vtmn. All the UK Biobank data used in the study are available at https://www.ukbiobank.ac.uk. The IMAGEN project data are available at https://imagen-project.org (*Schumann et al., 2010*). GWAS summary statistics used to calculate the PRS are available in the *Appendix 1— table 8*. Human gene expression data are available in the Allen Human Brain Atlas dataset: https://human.brainmap.org (*Hawrylycz et al., 2012*). Source data files have been provided for *Figures 2–5, 7 and 8*, which contain the numerical data used to generate the figures. Summary statistics of the GWAS for delayed brain development by Shi et al are available at https://delayedneurodevelopment.page.link/amTC.

The following dataset was generated:

| Author(s) | Year | Dataset title | Dataset URL | Database and Identifier |
|-----------|------|---------------|-------------|-------------------------|
| Duan H, Shi R, Kang J | 2024 | The summary statistics of GWAS for individual deviations of total GMV | https://doi.org/10.5061/dryad.jh9w0vtmn | Dryad, 10.5061/dryad.jh9w0vtmn |

The following previously published datasets were used:

| Author(s) | Year | Dataset title | Dataset URL | Database and Identifier |
|---|---|---|---|---|
| Nalls MA, Blauwendraat C, Vallerga CL | 2019 | Parkinson's disease or first degree relation to individual with Parkinson's disease | https://www.ebi.ac.uk/gwas/studies/GCST009325 | GWAS Catalog, GCST009325 |
| Sullivan P | 2021 | adhd2019 | https://doi.org/10.6084/m9.figshare.14671965 | figshare, 10.6084/m9.figshare.14671965 |
| Sullivan P | 2019 | asd2019 | https://doi.org/10.6084/m9.figshare.14671989 | figshare, 10.6084/m9.figshare.14671989 |
| Jansen IE, Savage JE, Watanabe K | 2019 | Summary statistics for Genome-wide meta-analysis identifies new loci and functional pathways influencing Alzheimer's disease risk | https://vu.data.surfsara.nl/index.php/s/l7aiRr1UEgdoJfZ | Vrije Universiteit Research Drive, l7aiRr1UEgdoJfZ |
| Sullivan P | 2023 | bip2021_noUKBB | https://doi.org/10.6084/m9.figshare.22564402 | figshare, 10.6084/m9.figshare.22564402 |
| Adams MJ | 2023 | MDD2 (MDD2018) GWAS sumstats w/o UKBB | https://doi.org/10.6084/m9.figshare.21655784 | figshare, 10.6084/m9.figshare.21655784 |
| Sullivan P | 2022 | scz2022 | https://doi.org/10.6084/m9.figshare.19426775 | figshare, 10.6084/m9.figshare.19426775 |

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

## Appendix 1

Population clustering of structural brain aging and its association with brain development.

## Supplementary method

### Cognitive assessment

#### Reaction time

This cognitive function test is based on 12 rounds of the card-game 'Snap'. The participant is shown two cards at a time; if both cards are the same, they press a button-box that is on the table in front of them as quickly as possible. The score used for analysis is mean time to correctly identify matches (UK Biobank data field 20023), which is the mean duration to first press of snap-button summed over rounds in which both cards matched.

#### Numeric memory

The participant was shown a 2-digit number to remember. The number then disappeared and after a short while they were asked to enter the number onto the screen. The number became one digit longer each time they remembered correctly (up to a maximum of 12 digits). This test is available for a subset of participants. The score used for analysis is maximum digits remembered correctly (UK Biobank data field 4282), which is longest number correctly recalled during the numeric memory test.

#### Fluid intelligence / reasoning

'Fluid intelligence' is defined as the capacity to solve problems that require logic and reasoning ability, independent of acquired knowledge. The participant has 2 min to complete as many questions as possible from the test. This test was incorporated into the touchscreen towards the end of recruitment. The score used for analysis is fluid intelligence score (UK Biobank data field 20016), which is a simple unweighted sum of the number of correct answers given to the 13 fluid intelligence questions. Participants who did not answer all of the questions within the allotted 2 min limit are scored as zero for each of the unattempted questions.

#### Trail making

The participant was presented with sets of digits/letters in circles scattered around the screen and asked to click on them sequentially according to a specific algorithm. The scores used for analysis are duration to complete numeric path (trail #1) (UK Biobank data field 6348) and duration to complete alphanumeric path (trail #2) (UK Biobank data field 6350).

#### Matrix pattern completion

The participant was presented with a series of matrix pattern blocks with an element missing and asked to select the element that best completed the pattern from a range of displayed choices. The score used for analysis is number of puzzles correctly solved (UK Biobank data field 6373).

#### Symbol digit substitution

The participant was presented with one grid linking symbols to single-digit integers and a second grid containing only the symbols. They were then asked to indicate the numbers attached to each of the symbols in the second grid using the first one as a key. The score used for analysis is number of symbol digit matches made correctly (UK Biobank data field 23324).

#### Tower rearranging

The participant was presented with an illustration of three pegs (towers) on which three differently-colored hoops had been placed. The were then asked to indicate how many moves it would take to re-arrange the hoops into another specific position. The score used for analysis is number of puzzles correct (UK Biobank data field 21004).

#### Paired associate learning

In the paired associate learning test the participants were shown 12 pairs of words (for 30 s in total) then, after an interval (in which they did a different test), presented with the first word of 10 of these

pairs and asked to select the matching second word from a choice of four alternatives. The words were presented in the order: huge, happy, tattered, old, long, red, sulking, pretty, tiny and new. The score used for analysis is number of word pairs correctly associated (UK Biobank data field 21097), which is the number of word pairs correctly associated out of 10 attempts.

## Prospective memory

Early in the touchscreen cognitive section, the participant is shown the message "At the end of the games we will show you four colored shapes and ask you to touch the Blue Square. However, to test your memory, we want you to actually touch the Orange Circle instead." The score used for analysis is prospective memory result (UK Biobank data field 20018), which condenses the results of the prospective memory test into 3 groups ("0": instruction not recalled, either skipped or incorrect, "1": correct recall on first attempt, "2": correct recall on second attempt). We divided the test results into two groups for simplicity of analysis: "0" indicating no correct recall, and the combination of "1" and "2" indicating correct recall.

## Pairs matching

Participants are asked to memorize the position of as many matching pairs of cards as possible. The cards are then turned face down on the screen and the participant is asked to touch as many pairs as possible in the fewest tries. Multiple rounds were conducted. The first round used 3 pairs of cards and the second 6 pairs of cards. In the pilot phase an additional (i.e. third) round was conducted using six pairs of cards. However this was dropped from the main study as the extra set of results were very similar to the second and not felt to add significant new information. The score used for analysis is number of incorrect matches in round 2 (UK Biobank data field 399.2).

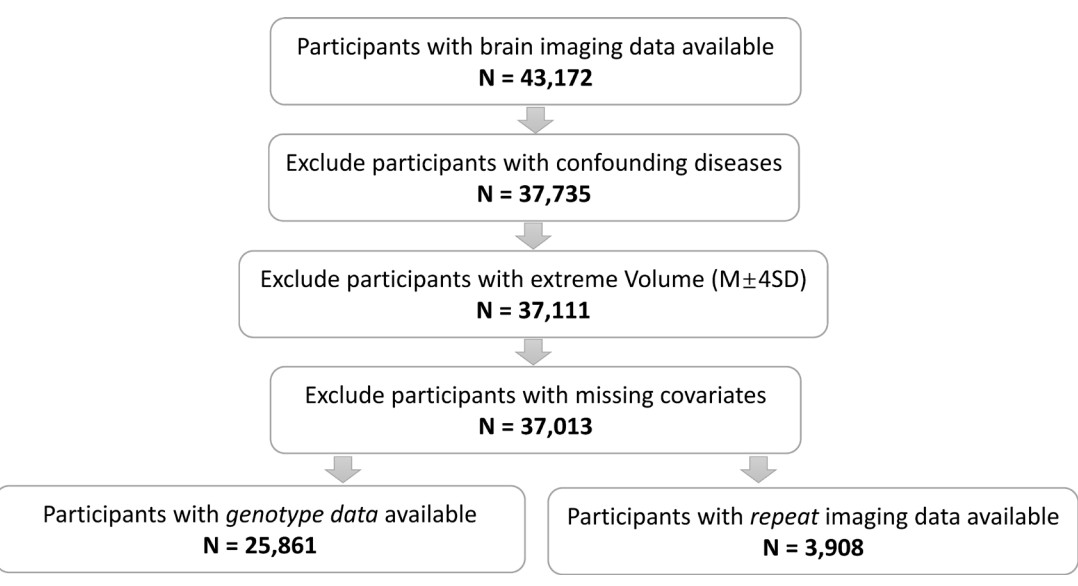

**Appendix 1—figure 1.** The sample selection workflow.

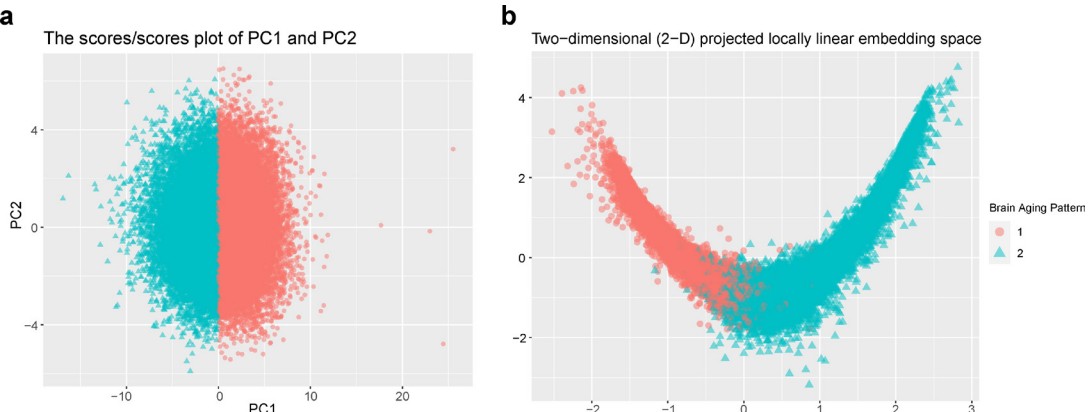

**Appendix 1—figure 2.** Estimated rates of change in regional volumes for 33 bilateral brain regions.

**a** The scores/scores plot of PC1 and PC2

**b** Two-dimensional (2–D) projected locally linear embedding space

Brain Aging Pattern
● 1
▲ 2

**Appendix 1—figure 3.** Stratification of the identified brain aging patterns using linear and non-linear dimensionality reduction methods.

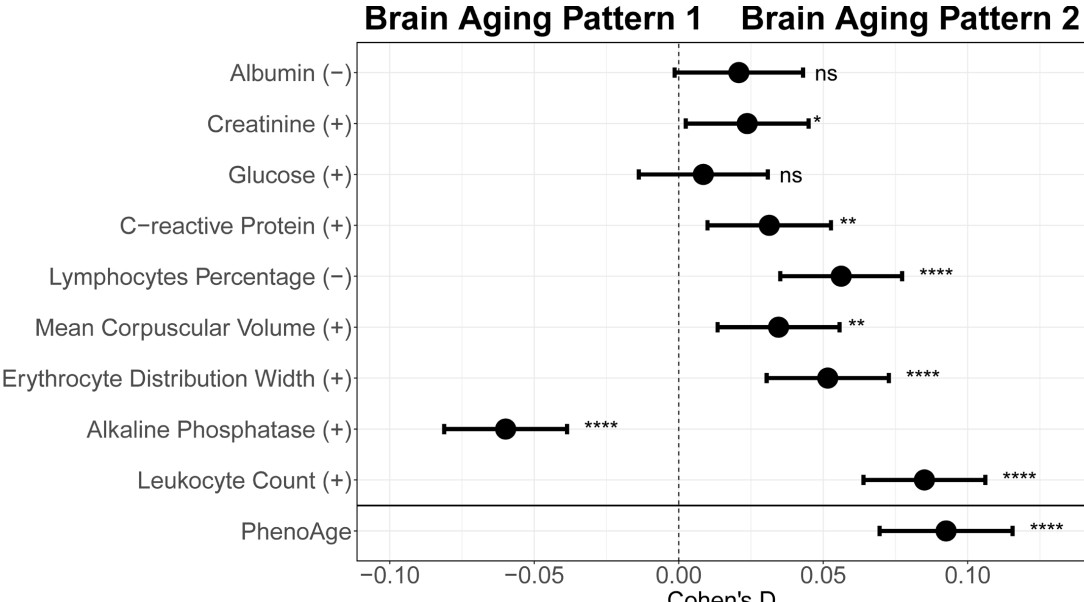

**Appendix 1—figure 4.** Effect size for comparing each individual blood biochemical metric (used to calculate the PhenoAge) between participants with brain aging patterns 1 and 2.

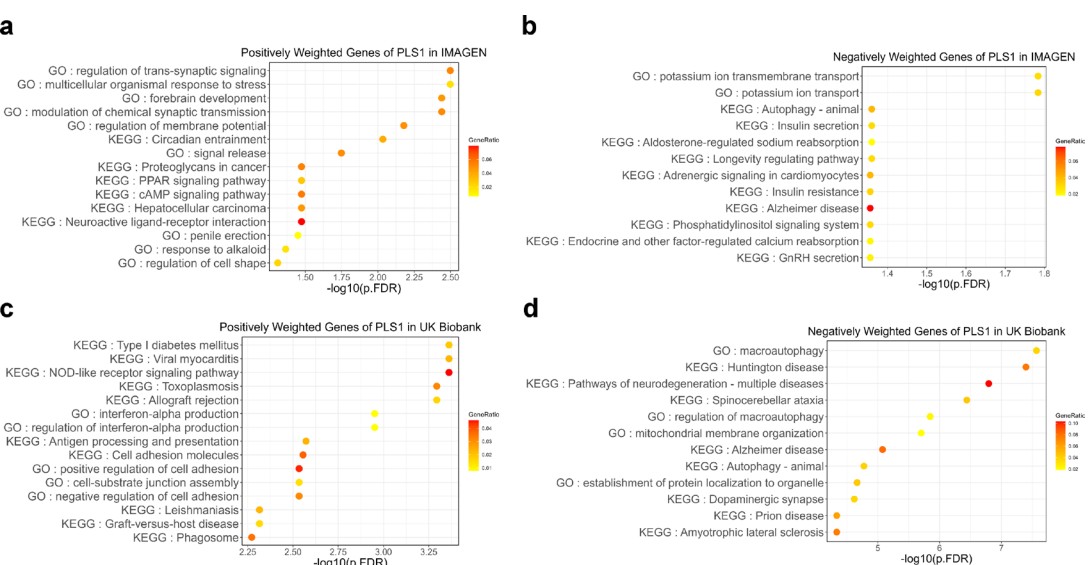

**Appendix 1—figure 5.** Gene set enrichment of Kyoto Encyclopedia of Genes and Genomes (KEGG) pathways and gene ontology (GO) of biological processes.

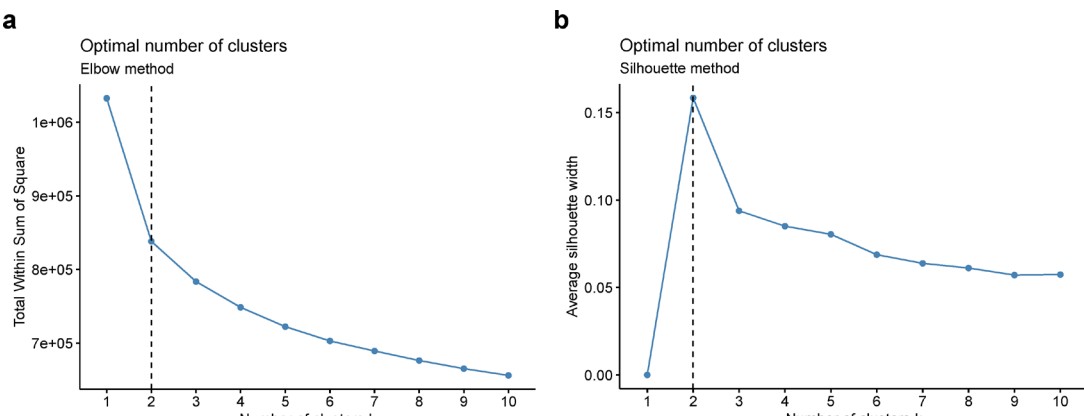

**Appendix 1—figure 6.** Optimal number of clusters was chosen using elbow method (**a**) and silhouette method (**b**).

**Appendix 1—table 1.** ICD-10 primary and secondary diagnostic codes for exclusion criteria.

| Condition | Code |
|---|---|
| Malignant neoplasm | C70, C71 |
| Dementia | F00, F01, F02, F03, F04 |
| Mental and behavioural disorders due to psychoactive substance use | F10-F19 |
| Schizophrenia, schizotypal and delusional disorders | F20-F29 |
| Mood [affective] disorders | F30-F39 |
| Mental retardation | F70-F79 |
| Disorders of psychological development | F80-F89 |
| Hyperkinetic disorders | F90 |
| Inflammatory diseases of the central nervous system | G00-G09 |
| Systemic atrophies primarily affecting the central nervous system | G10, G11, G122, G13 |
| Extrapyramidal and movement disorders | G20, G21, G22, G23 |
| Other degenerative diseases of the nervous system | G30-G32 |
| Demyelinating diseases of the central nervous system | G35-G37 |
| Episodic and paroxysmal disorders | G40, G41, G45, G46 |
| Infantile cerebral palsy | G80 |
| Cerebrovascular diseases | I60-I69 |
| Down's syndrome | Q90 |
| Intracranial injury | S06 |

**Appendix 1—table 2.** Self-reported illness codes for exclusion criteria.

| Field ID | Condition | Code |
|---|---|---|
| | Brain cancer/primary malignant brain tumour | 1032 |
| 20001 | Meningeal cancer/malignant meningioma | 1031 |

*Appendix 1—table 2 Continued on next page*

*Appendix 1—table 2 Continued*

| Field ID | Condition | Code |
|---|---|---|
| | Benign neuroma | 1683 |
| | Brain abscess/intracranial abscess | 1245 |
| | Brain haemorrhage | 1491 |
| | Cerebral aneurysm | 1425 |
| | Cerebral palsy | 1433 |
| | Chronic/degenerative neurological problem | 1258 |
| | dementia/alzheimers/cognitive impairment | 1263 |
| | Encephalitis | 1246 |
| | Epilepsy | 1264 |
| | Fracture skull/head | 1626 |
| | Head injury | 1266 |
| | Ischaemic stroke | 1583 |
| | Meningioma/benign meningeal tumour | 1659 |
| | Meningitis | 1247 |
| | Motor Neurone Disease | 1259 |
| | Multiple Sclerosis | 1261 |
| | Nervous system infection | 1244 |
| | Neurological injury/trauma | 1240 |
| | Other demyelinating disease (not Multiple Sclerosis) | 1397 |
| | Other neurological problem | 1434 |
| | Parkinson's Disease | 1262 |
| | Spina Bifida | 1524 |
| | Stroke | 1081 |
| | Subarachnoid haemorrhage | 1086 |
| | Subdural haemorrhage/haematoma | 1083 |
| 20002 | Transient ischaemic attack | 1082 |

**Appendix 1—table 3.** Cortical and subcortical brain regions.

**Desikan–Killiany Atlas**

bankssts
caudal anterior cingulate
caudal middle frontal
cuneus
entorhinal
fusiform
inferior parietal
inferior temporal
isthmus cingulate
lateral occipital
lateral orbitofrontal
lingual
medial orbitofrontal
middle temporal
parahippocampal
paracentral
pars opercularis
pars orbitalis
pars triangularis
pericalcarine
postcentral
posterior cingulate
precentral
precuneus
rostral anterior cingulate
rostral middle frontal
superior frontal
superior parietal
superior temporal
supramarginal
frontal pole
transverse temporal
insula

**ASEG Atlas**

thalamus proper
caudate
putamen
pallidum
hippocampus
amygdala
accumbens area

**Appendix 1—table 4.** Loadings matrix for the first 15 principal components.

| | PC1 | PC2 | PC3 | PC4 | PC5 | PC6 | PC7 | PC8 | PC9 | PC10 | PC11 | PC12 | PC13 | PC14 | PC15 |
|---|---|---|---|---|---|---|---|---|---|---|---|---|---|---|---|
| Cortical | | | | | | | | | | | | | | | |
| bankssts | 0.12 | –0.25 | 0.01 | 0.30 | –0.13 | 0.01 | 0.09 | 0.34 | 0.04 | 0.05 | –0.10 | 0.16 | –0.09 | –0.20 | 0.02 |
| caudal.anterior.cingulate | 0.12 | –0.06 | –0.02 | 0.13 | 0.39 | 0.28 | 0.13 | –0.09 | –0.37 | 0.24 | 0.05 | 0.03 | –0.04 | –0.13 | 0.07 |
| caudal.middle.frontal | 0.15 | 0.01 | –0.10 | –0.11 | 0.34 | –0.21 | 0.10 | 0.06 | 0.03 | –0.28 | –0.23 | –0.20 | 0.24 | 0.00 | 0.25 |
| cuneus | 0.13 | 0.47 | –0.01 | 0.06 | –0.01 | 0.00 | 0.07 | 0.12 | 0.01 | 0.07 | 0.12 | –0.05 | –0.09 | 0.08 | –0.07 |
| entorhinal | 0.09 | 0.07 | 0.13 | 0.08 | 0.10 | 0.17 | –0.53 | –0.18 | 0.06 | 0.04 | –0.19 | –0.10 | 0.13 | 0.01 | –0.04 |
| fusiform | 0.18 | –0.03 | 0.01 | 0.13 | –0.02 | 0.02 | –0.20 | –0.02 | 0.15 | 0.30 | –0.01 | –0.28 | –0.23 | 0.08 | 0.21 |
| inferior.parietal | 0.15 | –0.18 | 0.01 | 0.39 | –0.09 | 0.01 | 0.14 | –0.04 | 0.16 | 0.05 | –0.20 | –0.02 | –0.15 | –0.02 | 0.02 |
| inferior.temporal | 0.16 | –0.13 | 0.01 | 0.26 | –0.01 | 0.15 | –0.04 | –0.10 | 0.14 | –0.08 | –0.08 | –0.25 | 0.20 | 0.29 | –0.01 |
| isthmus.cingulate | 0.15 | 0.23 | 0.02 | 0.11 | –0.14 | –0.03 | 0.08 | –0.03 | –0.23 | –0.07 | –0.22 | 0.16 | 0.43 | 0.04 | 0.11 |
| lateral.occipital | 0.16 | 0.30 | 0.00 | 0.12 | –0.02 | 0.02 | 0.05 | 0.10 | 0.16 | 0.18 | 0.06 | –0.23 | –0.23 | 0.12 | 0.17 |
| lateral.orbitofrontal | 0.22 | –0.04 | –0.08 | –0.17 | –0.05 | 0.15 | –0.02 | –0.15 | 0.09 | –0.02 | –0.01 | 0.40 | –0.01 | 0.22 | 0.17 |
| lingual | 0.12 | 0.43 | 0.03 | 0.14 | –0.01 | –0.02 | 0.07 | 0.11 | –0.01 | –0.03 | –0.10 | 0.16 | 0.02 | –0.09 | –0.08 |

*Appendix 1—table 4 Continued on next page*

*Appendix 1—table 4 Continued*

| | PC1 | PC2 | PC3 | PC4 | PC5 | PC6 | PC7 | PC8 | PC9 | PC10 | PC11 | PC12 | PC13 | PC14 | PC15 |
|---|---|---|---|---|---|---|---|---|---|---|---|---|---|---|---|
| medial.orbitofrontal | 0.21 | –0.06 | –0.06 | –0.06 | –0.05 | 0.10 | –0.01 | –0.13 | 0.18 | –0.06 | 0.13 | 0.17 | 0.05 | 0.01 | –0.25 |
| middle.temporal | 0.17 | –0.17 | –0.01 | 0.27 | –0.14 | 0.10 | 0.21 | 0.25 | 0.09 | –0.15 | –0.09 | 0.04 | 0.07 | 0.05 | –0.11 |
| parahippocampal | 0.12 | 0.09 | 0.13 | 0.05 | 0.06 | 0.14 | –0.46 | 0.02 | –0.02 | 0.09 | –0.22 | 0.09 | 0.03 | –0.32 | 0.02 |
| paracentral | 0.19 | –0.05 | –0.10 | –0.07 | 0.18 | –0.23 | –0.11 | –0.06 | 0.01 | 0.03 | –0.02 | 0.06 | –0.35 | –0.01 | –0.35 |
| pars.opercularis | 0.16 | –0.02 | –0.06 | –0.23 | –0.19 | 0.10 | 0.14 | 0.01 | –0.16 | –0.13 | –0.43 | –0.29 | –0.16 | –0.06 | 0.00 |
| pars.orbitalis | 0.18 | 0.02 | –0.05 | –0.17 | –0.11 | 0.24 | 0.03 | –0.17 | 0.23 | 0.08 | 0.01 | 0.41 | –0.14 | 0.05 | 0.24 |
| pars.triangularis | 0.16 | 0.02 | –0.03 | –0.35 | –0.26 | 0.23 | 0.13 | –0.05 | –0.06 | –0.06 | –0.25 | –0.17 | –0.21 | –0.08 | –0.05 |
| pericalcarine | 0.10 | 0.47 | –0.01 | 0.08 | –0.01 | 0.01 | 0.09 | 0.13 | –0.02 | –0.01 | 0.02 | 0.08 | –0.03 | 0.03 | –0.17 |
| postcentral | 0.20 | –0.04 | –0.11 | 0.06 | 0.16 | –0.29 | –0.09 | 0.12 | 0.08 | 0.03 | –0.09 | 0.10 | 0.04 | 0.01 | 0.04 |
| posterior.cingulate | 0.18 | –0.08 | –0.04 | 0.02 | 0.24 | 0.09 | 0.08 | –0.02 | –0.30 | 0.01 | –0.07 | 0.10 | –0.05 | –0.12 | –0.31 |
| precentral | 0.21 | –0.01 | –0.10 | –0.08 | 0.28 | –0.29 | –0.09 | 0.08 | 0.15 | –0.05 | –0.15 | 0.13 | –0.06 | 0.05 | 0.12 |
| precuneus | 0.20 | 0.02 | –0.02 | 0.13 | –0.24 | –0.26 | 0.01 | –0.35 | –0.20 | –0.04 | 0.06 | –0.04 | 0.00 | 0.03 | 0.07 |
| rostral.anterior.cingulate | 0.15 | –0.07 | –0.07 | 0.05 | 0.31 | 0.21 | 0.15 | –0.05 | –0.30 | 0.13 | 0.11 | –0.06 | 0.00 | 0.18 | 0.15 |
| rostral.middle.frontal | 0.20 | 0.02 | –0.05 | –0.05 | 0.10 | 0.14 | 0.22 | –0.11 | 0.17 | –0.11 | 0.24 | –0.14 | 0.19 | –0.22 | 0.14 |
| superior.frontal | 0.22 | –0.03 | –0.09 | –0.21 | 0.14 | –0.13 | –0.01 | 0.06 | 0.20 | –0.18 | 0.06 | –0.11 | 0.02 | 0.00 | –0.16 |
| superior.parietal | 0.17 | 0.02 | –0.05 | 0.14 | –0.18 | –0.29 | 0.01 | –0.45 | –0.16 | 0.00 | 0.14 | –0.08 | –0.08 | –0.06 | 0.04 |
| superior.temporal | 0.21 | –0.13 | –0.02 | –0.06 | –0.10 | 0.01 | –0.07 | 0.40 | –0.07 | 0.13 | 0.24 | –0.05 | 0.02 | –0.06 | 0.08 |
| supramarginal | 0.17 | –0.15 | –0.08 | 0.06 | –0.24 | –0.23 | –0.13 | 0.03 | –0.21 | 0.04 | 0.11 | 0.10 | 0.14 | –0.08 | –0.06 |
| frontal.pole | 0.16 | 0.03 | –0.10 | 0.01 | 0.00 | 0.16 | 0.03 | –0.13 | 0.28 | 0.05 | 0.28 | –0.16 | 0.26 | –0.38 | –0.22 |
| transverse.temporal | 0.15 | –0.01 | –0.10 | –0.27 | –0.17 | –0.10 | –0.16 | 0.25 | –0.22 | 0.19 | 0.24 | –0.10 | 0.09 | 0.04 | 0.16 |
| insula | 0.19 | –0.08 | 0.01 | –0.22 | –0.12 | 0.10 | –0.07 | 0.11 | –0.06 | 0.15 | –0.08 | 0.00 | 0.19 | 0.16 | –0.23 |
| Subcortical | | | | | | | | | | | | | | | |
| thalamus.proper | 0.09 | –0.02 | 0.31 | –0.08 | 0.03 | –0.08 | 0.08 | 0.00 | 0.02 | –0.21 | 0.11 | 0.10 | –0.19 | –0.37 | 0.33 |
| caudate | 0.06 | –0.03 | 0.32 | –0.12 | 0.02 | –0.10 | 0.15 | –0.02 | 0.11 | 0.31 | –0.11 | 0.09 | 0.27 | 0.14 | 0.12 |
| putamen | 0.08 | –0.03 | 0.42 | –0.12 | 0.03 | –0.13 | 0.13 | –0.03 | 0.05 | 0.22 | –0.05 | –0.14 | 0.07 | –0.02 | –0.17 |
| pallidum | 0.03 | –0.03 | 0.42 | –0.07 | 0.00 | –0.13 | 0.16 | –0.06 | 0.04 | 0.20 | –0.07 | 0.00 | 0.04 | –0.20 | –0.06 |
| hippocampus | 0.12 | 0.01 | 0.33 | 0.04 | –0.02 | 0.11 | –0.20 | 0.08 | –0.16 | –0.39 | 0.09 | 0.03 | –0.11 | –0.03 | 0.05 |
| amygdala | 0.13 | –0.03 | 0.31 | 0.05 | –0.01 | 0.12 | –0.13 | 0.09 | –0.07 | –0.35 | 0.25 | –0.09 | –0.02 | 0.24 | –0.01 |
| accumbens.area | 0.11 | –0.05 | 0.31 | –0.02 | 0.12 | –0.07 | 0.11 | –0.04 | 0.00 | –0.03 | 0.10 | 0.09 | –0.14 | 0.34 | –0.18 |

**Appendix 1—table 5.** Baseline and demographic characteristics for participants in the total population and stratified by brain aging patterns.

| | Total (n=37,013) | Pattern 1 (n=18,929) | Pattern 2 (n=18,084) |
|---|---|---|---|
| Age (years), mean (SD) | 63.9 (7.63) | 63.9 (7.64) | 63.8 (7.63) |
| Female, n (%) | 19,958 (53.9) | 10,117 (53.4) | 9,841 (54.4) |
| Ethnicity, n (%) | | | |
| White | 34,219 (92.5) | 17,509 (92.5) | 16,710 (92.4) |
| Mixed | 1,137 (3.1) | 573 (3.0) | 564 (3.1) |
| Asian or Asian British | 1,210 (3.3) | 612 (3.2) | 598 (3.3) |
| Other | 447 (1.2) | 235 (1.2) | 212 (1.2) |

*Appendix 1—table 5 Continued on next page*

*Appendix 1—table 5 Continued*

|  | Total (n=37,013) | Pattern 1 (n=18,929) | Pattern 2 (n=18,084) |
|---|---|---|---|
| Smoking status, n (%)* |  |  |  |
| Never smoker | 23,633 (64.4) | 12,269 (65.4) | 11,364 (63.4) |
| Previous smoker | 12,213 (33.3) | 6,085 (32.4) | 6,128 (34.2) |
| Current smoker | 833 (2.3) | 414 (2.2) | 419 (2.3) |
| TDI, mean (SD)† | −1.94 (2.69) | −1.97 (2.66) | −1.90 (2.71) |
| BMI (kg/m²), mean (SD)‡ | 26.4 (4.32) | 26.3 (4.17) | 26.5 (4.46) |
| Years of Schooling, mean (SD)§ | 16.8 (4.32) | 16.9 (4.29) | 16.8 (4,35) |

TDI = Townsend Deprivation Index, BMI = Body Mass Index.

*Missing 334

†Missing 36

‡Missing 1937

§Missing 337

**Appendix 1—table 6.** Associations between results of biological aging biomarkers and subgroups stratified by whole-brain TGMV trajectories.

Cohen's d measures the standardized difference of means between brain aging pattern 2 and brain aging pattern 1.

| Biological aging biomarkers | Brain aging pattern 1 | | Brain aging pattern 2 | |
|---|---|---|---|---|
|  | N | Mean(SD) | N | Mean(SD) |
| LTL | 17,691 | 0.083 (0.98) | 16,876 | 0.055 (0.97) |
| PhenoAge | 15,228 | 41.35 (8.17) | 14,323 | 41.58 (8.32) |

| Biological aging biomarkers | Unadjusted | | | Adjusted | | |
|---|---|---|---|---|---|---|
|  | Cohen's d (95% CI) | p | P.Bonferroni | Cohen's d (95% CI) | p | P.Bonferroni |
| LTL | −0.028 (-0.049,−0.007) | 0.009 | 0 | −0.030 (-0.051,−0.009) | 0.006 | 0.011 |
| PhenoAge | 0.027 (0.004, 0.050) | 0.019 | 0 | 0.092 (0.070, 0.116) | 3.05E-15 | 6.11E-15 |

**Appendix 1—table 7.** Associations between results of cognitive function tests and subgroups stratified by whole-brain TGMV trajectories.

Cohen's d measures the standardized difference of means between brain aging pattern 2 and brain aging pattern 1.

| Cognitive functions | Brain aging pattern 1 | | Brain aging pattern 2 | |
|---|---|---|---|---|
|  | N | Mean(SD) | N | Mean(SD) |
| Reaction time | 17,749 | −594.70 (108.15) | 16,831 | −594.68 (110.50) |
| Numeric memory | 13,350 | 6.82 (1.26) | 12,346 | 6.72 (1.27) |
| Fluid intelligence | 17,580 | 6.73 (2.06) | 16,612 | 6.53 (2.04) |
| Trail making A | 13,052 | −224.50 (84.82) | 12,036 | −227.56 (84.85) |
| Trail making B | 12,743 | −563.06 (246.74) | 11,753 | −576.55 (260.75) |
| Matrix pattern completion | 13,064 | 8.07 (2.11) | 12,064 | 7.91 (2.14) |
| Symbol digit substitution | 13,077 | 19.08 (5.15) | 12,056 | 18.87 (5.35) |
| Tower rearranging | 12,952 | 9.96 (3.20) | 11,958 | 9.83 (3.23) |

*Appendix 1—table 7 Continued on next page*

*Appendix 1—table 7 Continued*

| Cognitive functions | Brain aging pattern 1 | | Brain aging pattern 2 | |
|---|---|---|---|---|
| | N | Mean(SD) | N | Mean(SD) |
| Paired associate learning | 13,184 | 7.01 (2.59) | 12,198 | 6.88 (2.65) |
| Prospective memory | 17,831 | N/A | 16,949 | N/A |
| Pairs matching | 17,840 | −3.58 (2.90) | 16,956 | −3.67 (2.94) |

| Cognitive functions | Unadjusted | | | Adjusted | | |
|---|---|---|---|---|---|---|
| | Cohen's d (95% CI) | P | P.FDR | Cohen's d (95% CI) | P | P.FDR |
| Reaction time | 0.000 (-0.021, 0.021) | 0.99 | 0.99 | 0.006 (-0.016, 0.028) | 0.61 | 0.61 |
| Numeric memory | −0.082 (-0.106,−0.057) | 5.97E-11 | 3.28E-10 | −0.080 (-0.106,−0.055) | 8.99E-10 | 4.95E-09 |
| Fluid intelligence | −0.102 (-0.123,−0.080) | 5.94E-21 | 6.54E-20 | −0.99 (-0.121,−0.077) | 3.30E-18 | 3.63E-17 |
| Trail making A | −0.036 (-0.061,−0.011) | 0.004 | 0.006 | −0.050 (-0.074,−0.024) | 1.46E-04 | 2.29E-04 |
| Trail making B | −0.053 (-0.078,−0.028) | 3.16E-05 | 8.68E-05 | −0.067 (-0.093,−0.041) | 6.16E-07 | 1.69E-06 |
| Matrix pattern completion | −0.076 (-0.101,−0.051) | 1.84E-09 | 6.74E-09 | −0.078 (-0.104,−0.052) | 3.67E-09 | 1.35E-08 |
| Symbol digit substitution | −0.040 (-0.065,−0.015) | 0.002 | 0.002 | −0.053 (-0.079,−0.027) | 6.15E-05 | 1.13E-04 |
| Tower rearranging | −0.041 (-0.066,−0.016) | 0.001 | 0.002 | −0.049 (-0.075,−0.023) | 2.18E-04 | 3.00E-04 |
| Paired associate learning | −0.051 (-0.076,−0.027) | 4.64E-05 | 1.02E-04 | −0.054 (-0.079,−0.028) | 4.89E-05 | 1.08E-04 |
| Prospective memory | OR: 0.943 (0.891, 0.999) | 0.047 | 0.052 | OR: 0.940 (0.883, 1.000) | 0.052 | 0.057 |
| Pairs matching | −0.029 (-0.050,−0.008) | 0.006 | 0.008 | −0.033 (-0.055,−0.011) | 0.003 | 0.004 |

**Appendix 1—table 8.** Genome-wide association study details.

Loci associated with risk were thresholded at $p<5×10^{-8}$, then distance-based clumping was used to define independently significant loci.

| Study | Number cases | Number controls | Number of genome-wide independently significant loci | Download link |
|---|---|---|---|---|
| Attention deficit hyperactivity disorder *Demontis et al., 2019* | 20,183 | 35,191 | 12 | https://figshare.com/ndownloader/files/28169253 |
| Autism spectrum disorder *Grove et al., 2019* | 18,381 | 27,969 | 5 | https://figshare.com/ndownloader/files/28169292 |
| Alzheimer's disease *Jansen et al., 2019* | 71,880 | 383,378 | 25 | https://ctg.cncr.nl/software/summary_statistics |
| Parkinson's disease *Nalls et al., 2019* | 37,688, and 18,618 (proxy-cases) | 1,417,791 | 90 | https://drive.google.com/file/d/1FZ9UL99LAqyWnyNBxxlx6qOUlfAnublN/view?usp=sharing |

*Appendix 1—table 8 Continued on next page*

*Appendix 1—table 8 Continued*

| Study | Number cases | Number controls | Number of genome-wide independently significant loci | Download link |
|---|---|---|---|---|
| Bipolar disorder *Mullins et al., 2021* | 41,917 | 371,549 | 64 | https://figshare.com/ndownloader/files/40036705 |
| Major depressive disorder *Wray et al., 2018* | 135,458 | 344,901 | 44 | https://figshare.com/ndownloader/files/39504667 |
| Schizophrenia *Trubetskoy et al., 2022* | 76,755 | 243,649 | 287 | https://figshare.com/ndownloader/files/34517828 |
| Delayed Brain Development *Shi et al., 2023* | 7662 (proxy phenotype, continuous) | | 1 | https://delayedneurodevelopment.page.link/amTC |

**Appendix 1—table 9.** Polygenic Risk Scores comparisons between two subgroups.
Data supporting these scores were obtained either entirely from external GWAS data (the Standard PRS set). The bold P values reflect significance after FDR correction.

| Trait | n1 | n2 | statistic | p | p.adjust |
|---|---|---|---|---|---|
| AAM | 18,429 | 17,586 | −2.218 | 0.027 | 0.080 |
| AMD | 18,429 | 17,586 | 1.753 | 0.080 | 0.169 |
| AD | 18,429 | 17,586 | 0.735 | 0.462 | 0.616 |
| AST | 18,429 | 17,586 | −0.861 | 0.389 | 0.543 |
| AF | 18,429 | 17,586 | −0.100 | 0.920 | 0.945 |
| BD | 18,429 | 17,586 | 3.557 | 3.75E-04 | 0.002 |
| BMI | 18,429 | 17,586 | −3.309 | 0.001 | 0.005 |
| CRC | 18,429 | 17,586 | −0.544 | 0.586 | 0.703 |
| BC | 18,429 | 17,586 | −3.140 | 0.002 | 0.008 |
| CVD | 18,429 | 17,586 | −2.104 | 0.035 | 0.091 |
| CED | 18,429 | 17,586 | 1.046 | 0.296 | 0.484 |
| CAD | 18,429 | 17,586 | −1.588 | 0.112 | 0.202 |
| CD | 18,429 | 17,586 | −0.094 | 0.925 | 0.945 |
| EOC | 18,429 | 17,586 | −2.183 | 0.029 | 0.080 |
| EBMDT | 18,429 | 17,586 | −11.343 | <1.00E-20 | <1.00E-20 |
| HBA1C_DF | 18,429 | 17,586 | −2.948 | 0.003 | 0.013 |
| HEIGHT | 18,429 | 17,586 | 6.658 | 2.81E-11 | 3.37E-10 |
| HDL | 18,429 | 17,586 | 0.884 | 0.377 | 0.543 |
| HT | 18,429 | 17,586 | −3.539 | 4.02E-04 | 0.002 |
| IOP | 18,429 | 17,586 | −1.605 | 0.109 | 0.202 |
| ISS | 18,429 | 17,586 | −2.383 | 0.017 | 0.056 |
| LDL_SF | 18,429 | 17,586 | −0.686 | 0.492 | 0.627 |
| MEL | 18,429 | 17,586 | 2.025 | 0.043 | 0.103 |
| MS | 18,429 | 17,586 | −0.069 | 0.945 | 0.945 |
| OP | 18,429 | 17,586 | 12.029 | <1.00E-20 | <1.00E-20 |

*Appendix 1—table 9 Continued*

| Trait | n1 | n2 | statistic | p | p.adjust |
|---|---|---|---|---|---|
| PD | 18,429 | 17,586 | 1.456 | 0.145 | 0.249 |
| POAG | 18,429 | 17,586 | −0.856 | 0.392 | 0.543 |
| PC | 18,429 | 17,586 | −0.240 | 0.810 | 0.941 |
| PSO | 18,429 | 17,586 | −1.781 | 0.075 | 0.169 |
| RA | 18,429 | 17,586 | −2.437 | 0.015 | 0.053 |
| SCZ | 18,429 | 17,586 | 0.158 | 0.874 | 0.945 |
| SLE | 18,429 | 17,586 | 1.695 | 0.090 | 0.180 |
| T1D | 18,429 | 17,586 | 0.666 | 0.505 | 0.627 |
| T2D | 18,429 | 17,586 | −5.523 | 3.35E-08 | 3.02E-07 |
| UC | 18,429 | 17,586 | 0.883 | 0.377 | 0.543 |
| VTE | 18,429 | 17,586 | −0.170 | 0.865 | 0.945 |

**Appendix 1—table 10.** Polygenic Risk Scores comparisons between two subgroups. Data supporting these scores were obtained external and internal UK Biobank data (the Enhanced PRS set). The bold p values reflect significance after FDR correction.

| Trait | n1 | n2 | statistic | p | p.adjust |
|---|---|---|---|---|---|
| AAM | 3,407 | 3,409 | −1.708 | 0.088 | 0.344 |
| AMD | 3,407 | 3,409 | 0.547 | 0.584 | 0.931 |
| AD | 3,407 | 3,409 | 0.756 | 0.450 | 0.820 |
| APOEA | 3,407 | 3,409 | 0.023 | 0.982 | 0.993 |
| APOEB | 3,407 | 3,409 | 0.119 | 0.905 | 0.968 |
| AST | 3,407 | 3,409 | 0.112 | 0.911 | 0.968 |
| AF | 3,407 | 3,409 | 1.306 | 0.192 | 0.600 |
| BD | 3,407 | 3,409 | 0.561 | 0.575 | 0.931 |
| BMI | 3,407 | 3,409 | −0.976 | 0.329 | 0.730 |
| CRC | 3,407 | 3,409 | 0.984 | 0.325 | 0.730 |
| BC | 3,407 | 3,409 | −0.995 | 0.320 | 0.730 |
| CAL | 3,407 | 3,409 | −1.786 | 0.074 | 0.326 |
| CVD | 3,407 | 3,409 | −0.009 | 0.993 | 0.993 |
| CED | 3,407 | 3,409 | 1.280 | 0.200 | 0.600 |
| CAD | 3,407 | 3,409 | 0.231 | 0.818 | 0.961 |
| DOA | 3,407 | 3,409 | −0.326 | 0.745 | 0.961 |
| EOC | 3,407 | 3,409 | −2.167 | 0.030 | 0.155 |
| EBMDT | 3,407 | 3,409 | −6.111 | 1.04E-09 | 2.65E-08 |
| EGCR | 3,407 | 3,409 | 0.413 | 0.680 | 0.961 |
| EGCY | 3,407 | 3,409 | −0.210 | 0.834 | 0.961 |
| HBA1C_DF | 3,407 | 3,409 | 0.130 | 0.896 | 0.968 |
| HEIGHT | 3,407 | 3,409 | 4.351 | 1.38E-05 | 2.35E-04 |
| HDL | 3,407 | 3,409 | 0.294 | 0.769 | 0.961 |
| HT | 3,407 | 3,409 | −0.884 | 0.377 | 0.743 |

*Appendix 1—table 10 Continued on next page*

*Appendix 1—table 10 Continued*

| Trait | n1 | n2 | statistic | p | p.adjust |
|-------|------|------|-----------|---------|----------|
| IOP | 3,407 | 3,409 | −2.366 | 0.018 | 0.151 |
| ISS | 3,407 | 3,409 | 0.066 | 0.947 | 0.986 |
| LDL_SF | 3,407 | 3,409 | −0.193 | 0.847 | 0.961 |
| MEL | 3,407 | 3,409 | 2.659 | 0.008 | 0.080 |
| MS | 3,407 | 3,409 | −2.293 | 0.022 | 0.151 |
| OTFA | 3,407 | 3,409 | 0.318 | 0.750 | 0.961 |
| OSFA | 3,407 | 3,409 | 0.770 | 0.441 | 0.820 |
| OP | 3,407 | 3,409 | 6.484 | 9.54E-11 | 4.87E-09 |
| PD | 3,407 | 3,409 | 1.041 | 0.298 | 0.730 |
| PDCL | 3,407 | 3,409 | 0.663 | 0.507 | 0.862 |
| PHG | 3,407 | 3,409 | 0.392 | 0.695 | 0.961 |
| PFA | 3,407 | 3,409 | 0.495 | 0.621 | 0.932 |
| POAG | 3,407 | 3,409 | −1.083 | 0.279 | 0.730 |
| PC | 3,407 | 3,409 | 0.675 | 0.500 | 0.862 |
| PSO | 3,407 | 3,409 | −2.234 | 0.026 | 0.151 |
| RMNC | 3,407 | 3,409 | 0.501 | 0.617 | 0.932 |
| RHR | 3,407 | 3,409 | −2.865 | 0.004 | 0.053 |
| RA | 3,407 | 3,409 | 0.297 | 0.766 | 0.961 |
| SCZ | 3,407 | 3,409 | −0.880 | 0.379 | 0.743 |
| SGM | 3,407 | 3,409 | −0.224 | 0.823 | 0.961 |
| SLE | 3,407 | 3,409 | 1.458 | 0.145 | 0.493 |
| TCH | 3,407 | 3,409 | 0.191 | 0.848 | 0.961 |
| TFA | 3,407 | 3,409 | 0.892 | 0.372 | 0.743 |
| TTG | 3,407 | 3,409 | 1.212 | 0.226 | 0.640 |
| T1D | 3,407 | 3,409 | 1.771 | 0.077 | 0.326 |
| T2D | 3,407 | 3,409 | −2.218 | 0.027 | 0.151 |
| VTE | 3,407 | 3,409 | −1.613 | 0.107 | 0.390 |

**Appendix 1—table 11.** Most significant single-variant associations ($p < 5\,10^{-8}$) detected in the GWAS analyses.

Six independent SNPs at genome-wide significance level were identified by linkage disequilibrium (LD) clumping (r2 < 0.1 within a 250 kb window). The location (chromosome [chr] and base position [bp]), alleles (A1 = effect allele and A2 = other allele), effect (β) and its standard error (β SE) with respect to A1, and association p-values from regression model of the variants are given, along with functional consequences of SNPs on gene by performing ANNOVAR.

| SNP | A1 | A2 | p-value | β | β SE | Location (chr:bp) | Gene symbol | Position relative to gene |
|-----|-----|-----|---------|---|------|-------------------|-------------|---------------------------|
| rs10835187 | C | T | 1.70e-14 | −0.02558 | 0.003333 | 11:27505677 | LGR4, LIN7C | intergenic |
| rs7776725 | C | T | 4.47e-13 | −0.02640 | 0.003644 | 7:121033121 | FAM3C | intronic |
| rs779233904 | AAC | A | 1.57e-09 | 0.02083 | 0.003449 | 6:151910404 | CCDC170 | intronic |
| rs2504071 | T | C | 3.34e-09 | 0.01959 | 0.003311 | 6:152084862 | ESR1 | intronic |
| 17:43553496:A:AAT | A | AAT | 6.65e-09 | −0.02472 | 0.004261 | 17:43553496 | PLEKHM1 | intronic |

*Appendix 1—table 11 Continued on next page*

*Appendix 1—table 11 Continued*

| SNP | A1 | A2 | p-value | β | β SE | Location (chr:bp) | Gene symbol | Position relative to gene |
|---|---|---|---|---|---|---|---|---|
| 10:104227791:G:GA | GA | G | 1.48e-08 | –0.01889 | 0.003334 | 10:104227791 | TMEM180 | intronic |

**Appendix 1—table 12.** Association between gene expression profiles of mapped genes and estimated APC during brain development.

The bold p values reflect significance after the spatial permutation test.

| Gene Symbol | Spearman's ρ | p value | P.permutation |
|---|---|---|---|
| ACTR1A | 0.096 | 0.440 | 0.328 |
| ARHGAP27 | 0.051 | 0.684 | 0.445 |
| ARL17B | 0.047 | 0.705 | 0.452 |
| BDNF-AS | 0.256 | 0.038 | 0.038 |
| CCDC170 | 0.268 | 0.030 | 0.069 |
| ESR1 | 0.021 | 0.870 | 0.483 |
| FAM3C | –0.096 | 0.444 | 0.356 |
| KANSL1 | –0.262 | 0.034 | 0.073 |
| KANSL1-AS1 | 0.067 | 0.594 | 0.313 |
| LGR4 | 0.558 | 1.78E-06 | 2.50E-04 |
| LIN7C | 0.036 | 0.775 | 0.464 |
| LRRC37A4P | –0.272 | 0.027 | 0.148 |
| MAPT | 0.024 | 0.846 | 0.405 |
| PLEKHM1 | –0.276 | 0.025 | 0.109 |
| SPPL2C | 0.116 | 0.351 | 0.189 |
| STH | –0.147 | 0.238 | 0.277 |
| SUFU | 0.407 | 0.001 | 0.028 |

**Appendix 1—table 13.** Association between gene expression profiles of mapped genes and estimated APC during brain aging.

The bold p values reflect significance after the spatial permutation test.

| Gene Symbol | Spearman's ρ | Pvalue | P.permutation |
|---|---|---|---|
| ACTR1A | –0.235 | 0.058 | 0.052 |
| ARHGAP27 | 0.486 | 4.48E-05 | 5.50E-04 |
| ARL17B | 0.090 | 0.473 | 0.240 |
| BDNF-AS | 0.490 | 3.68E-05 | 1.50E-04 |
| CCDC170 | 0.206 | 0.098 | 0.075 |
| ESR1 | 0.532 | 6.02E-06 | 1.50E-04 |
| FAM3C | –0.366 | 0.003 | 0.005 |
| KANSL1 | 0.213 | 0.086 | 0.078 |
| KANSL1-AS1 | –0.262 | 0.034 | 0.059 |
| LGR4 | 0.070 | 0.576 | 0.348 |
| LIN7C | 0.177 | 0.154 | 0.120 |
| LRRC37A4P | 0.143 | 0.250 | 0.165 |

*Appendix 1—table 13 Continued on next page*

*Appendix 1—table 13 Continued*

| Gene Symbol | Spearman's ρ | Pvalue | P.permutation |
|---|---|---|---|
| MAPT | −0.287 | <u>0.020</u> | 0.022 |
| PLEKHM1 | 0.202 | 0.104 | 0.080 |
| SPPL2C | 0.211 | 0.089 | 0.059 |
| STH | −0.001 | 0.997 | 0.490 |
| SUFU | −0.036 | 0.773 | 0.373 |

**Appendix 1—table 14.** Model evaluation results using relative measures: AIC, BIC, likelihood ratio test and intra-class correlation (ICC).

| model | AIC | BIC | lrtest | ICC_adjusted | ICC_unadjusted |
|---|---|---|---|---|---|
| lmer_intr_thalamus.proper | 79444.24 | 79591.29 | 6.45E-15 | 0.8875 | 0.3958 |
| lmer_slope_thalamus.proper | 79382.89 | 79547.24 | 6.45E-15 | 0.8889 | 0.3986 |
| lmer_intr_caudate | 95845.48 | 95992.54 | 2.68E-71 | 0.9543 | 0.6824 |
| lmer_slope_caudate | 95524.49 | 95688.84 | 2.68E-71 | 0.9564 | 0.6817 |
| lmer_intr_putamen | 92106.06 | 92253.12 | 2.40E-42 | 0.9398 | 0.5946 |
| lmer_slope_putamen | 91918.4 | 92082.75 | 2.40E-42 | 0.9424 | 0.5933 |
| lmer_intr_pallidum | 90500.3 | 90647.36 | 4.21E-42 | 0.8859 | 0.5116 |
| lmer_slope_pallidum | 90313.76 | 90478.12 | 4.21E-42 | 0.8862 | 0.5093 |
| lmer_intr_hippocampus | 89833.92 | 89980.97 | 1.37E-08 | 0.9309 | 0.5505 |
| lmer_slope_hippocampus | 89801.71 | 89966.06 | 1.37E-08 | 0.9329 | 0.5505 |
| lmer_intr_amygdala | 89976.98 | 90124.03 | 4.01E-06 | 0.8697 | 0.4898 |
| lmer_slope_amygdala | 89956.13 | 90120.48 | 4.01E-06 | 0.8713 | 0.4897 |
| lmer_intr_accumbens.area | 96936.19 | 97083.24 | 1.69E-14 | 0.8228 | 0.5295 |
| lmer_slope_accumbens.area | 96876.77 | 97041.12 | 1.69E-14 | 0.8233 | 0.5310 |
| lmer_intr_bankssts | 97981.96 | 98129.01 | 0.001818 | 0.9339 | 0.6798 |
| lmer_slope_bankssts | 97973.34 | 98137.69 | 0.001818 | 0.9354 | 0.6814 |
| lmer_intr_caudal.anterior.cingulate | 109133.2 | 109280.3 | 0.131507 | 0.8638 | 0.7653 |
| lmer_slope_caudal.anterior.cingulate | 109133.2 | 109297.5 | 0.131507 | 0.8644 | 0.7659 |
| lmer_intr_caudal.middle.frontal | 93118.13 | 93265.19 | 8.31E-06 | 0.9227 | 0.5929 |
| lmer_slope_caudal.middle.frontal | 93098.74 | 93263.09 | 8.31E-06 | 0.9246 | 0.5949 |
| lmer_intr_cuneus | 101801.4 | 101948.5 | 0.014429 | 0.9242 | 0.7256 |
| lmer_slope_cuneus | 101796.9 | 101961.3 | 0.014429 | 0.9245 | 0.7262 |
| lmer_intr_entorhinal | 109404.9 | 109551.9 | 1.63E-14 | 0.8013 | 0.6908 |
| lmer_slope_entorhinal | 109345.4 | 109509.7 | 1.63E-14 | 0.8037 | 0.6928 |
| lmer_intr_fusiform | 87545.88 | 87692.93 | 9.17E-05 | 0.9139 | 0.5062 |
| lmer_slope_fusiform | 87531.28 | 87695.63 | 9.17E-05 | 0.9163 | 0.5074 |
| lmer_intr_inferior.parietal | 89044.43 | 89191.48 | 5.89E-14 | 0.9374 | 0.5564 |
| lmer_slope_inferior.parietal | 88987.51 | 89151.86 | 5.89E-14 | 0.9419 | 0.5603 |
| lmer_intr_inferior.temporal | 84066.47 | 84213.52 | 5.73E-07 | 0.9384 | 0.4956 |
| lmer_slope_inferior.temporal | 84041.73 | 84206.08 | 5.73E-07 | 0.9405 | 0.4977 |
| lmer_intr_isthmus.cingulate | 92442.12 | 92589.17 | 0.127191 | 0.9275 | 0.5862 |

*Appendix 1—table 14 Continued on next page*

*Appendix 1—table 14 Continued*

| model | AIC | BIC | lrtest | ICC_adjusted | ICC_unadjusted |
|---|---|---|---|---|---|
| lmer_slope_isthmus.cingulate | 92442 | 92606.35 | 0.127191 | 0.9284 | 0.5869 |
| lmer_intr_lateral.occipital | 89550.4 | 89697.45 | 0.003943 | 0.9121 | 0.5273 |
| lmer_slope_lateral.occipital | 89543.33 | 89707.68 | 0.003943 | 0.9129 | 0.5282 |
| lmer_intr_lateral.orbitofrontal | 86224.13 | 86371.18 | 1.29E-08 | 0.8466 | 0.4323 |
| lmer_slope_lateral.orbitofrontal | 86191.79 | 86356.14 | 1.29E-08 | 0.8497 | 0.4345 |
| lmer_intr_lingual | 102605.2 | 102752.2 | 0.041242 | 0.9181 | 0.7268 |
| lmer_slope_lingual | 102602.8 | 102767.2 | 0.041242 | 0.9181 | 0.7272 |
| lmer_intr_medial.orbitofrontal | 89954.36 | 90101.41 | 2.25E-06 | 0.7832 | 0.4219 |
| lmer_slope_medial.orbitofrontal | 89932.35 | 90096.7 | 2.25E-06 | 0.7872 | 0.4241 |
| lmer_intr_middle.temporal | 83331.01 | 83478.06 | 0.0019 | 0.9177 | 0.4615 |
| lmer_slope_middle.temporal | 83322.48 | 83486.83 | 0.0019 | 0.9195 | 0.4629 |
| lmer_intr_parahippocampal | 108997.1 | 109144.2 | 4.99E-05 | 0.8686 | 0.7639 |
| lmer_slope_parahippocampal | 108981.3 | 109145.6 | 4.99E-05 | 0.8690 | 0.7639 |
| lmer_intr_paracentral | 98058.63 | 98205.68 | 0.015686 | 0.8695 | 0.5958 |
| lmer_slope_paracentral | 98054.32 | 98218.67 | 0.015686 | 0.8705 | 0.5960 |
| lmer_intr_pars.opercularis | 96829.05 | 96976.1 | 1.20E-12 | 0.9354 | 0.6658 |
| lmer_slope_pars.opercularis | 96778.15 | 96942.5 | 1.20E-12 | 0.9396 | 0.6698 |
| lmer_intr_pars.orbitalis | 96989.61 | 97136.67 | 1.39E-05 | 0.8785 | 0.5892 |
| lmer_slope_pars.orbitalis | 96971.25 | 97135.6 | 1.39E-05 | 0.8805 | 0.5912 |
| lmer_intr_pars.triangularis | 96637.58 | 96784.63 | 4.55E-18 | 0.9402 | 0.6710 |
| lmer_slope_pars.triangularis | 96561.72 | 96726.07 | 4.55E-18 | 0.9439 | 0.6748 |
| lmer_intr_pericalcarine | 105115.9 | 105263 | 0.022841 | 0.9429 | 0.8190 |
| lmer_slope_pericalcarine | 105112.4 | 105276.7 | 0.022841 | 0.9441 | 0.8200 |
| lmer_intr_postcentral | 91605.6 | 91752.65 | 0.000549 | 0.8764 | 0.5189 |
| lmer_slope_postcentral | 91594.59 | 91758.94 | 0.000549 | 0.8789 | 0.5203 |
| lmer_intr_posterior.cingulate | 96853.73 | 97000.78 | 2.21E-19 | 0.8913 | 0.6014 |
| lmer_slope_posterior.cingulate | 96771.82 | 96936.17 | 2.21E-19 | 0.8949 | 0.6022 |
| lmer_intr_precentral | 91186.59 | 91333.64 | 2.61E-12 | 0.8478 | 0.4864 |
| lmer_slope_precentral | 91137.25 | 91301.6 | 2.61E-12 | 0.8504 | 0.4864 |
| lmer_intr_precuneus | 84734.58 | 84881.63 | 1.23E-08 | 0.9088 | 0.4708 |
| lmer_slope_precuneus | 84702.16 | 84866.51 | 1.23E-08 | 0.9126 | 0.4730 |
| lmer_intr_rostral.anterior.cingulate | 95688.68 | 95835.73 | 2.64E-12 | 0.9093 | 0.6097 |
| lmer_slope_rostral.anterior.cingulate | 95639.36 | 95803.72 | 2.64E-12 | 0.9122 | 0.6129 |
| lmer_intr_rostral.middle.frontal | 80873.84 | 81020.89 | 2.57E-17 | 0.9137 | 0.4354 |
| lmer_slope_rostral.middle.frontal | 80801.44 | 80965.79 | 2.57E-17 | 0.9191 | 0.4395 |
| lmer_intr_superior.frontal | 79730.6 | 79877.65 | 1.25E-11 | 0.8921 | 0.4038 |
| lmer_slope_superior.frontal | 79684.38 | 79848.73 | 1.25E-11 | 0.8972 | 0.4060 |
| lmer_intr_superior.parietal | 92895.95 | 93043 | 8.55E-07 | 0.8899 | 0.5487 |

*Appendix 1—table 14 Continued on next page*

*Appendix 1—table 14 Continued*

| model | AIC | BIC | lrtest | ICC_adjusted | ICC_unadjusted |
|---|---|---|---|---|---|
| lmer_slope_superior.parietal | 92872.01 | 93036.36 | 8.55E-07 | 0.8934 | 0.5512 |
| lmer_intr_superior.temporal | 86495.24 | 86642.29 | 0.003021 | 0.9148 | 0.4959 |
| lmer_slope_superior.temporal | 86487.64 | 86651.99 | 0.003021 | 0.9168 | 0.4971 |
| lmer_intr_supramarginal | 87390.39 | 87537.44 | 3.67E-13 | 0.9263 | 0.5221 |
| lmer_slope_supramarginal | 87337.12 | 87501.47 | 3.67E-13 | 0.9320 | 0.5258 |
| lmer_intr_frontal.pole | 110426.3 | 110573.4 | 1.95E-65 | 0.6907 | 0.5868 |
| lmer_slope_frontal.pole | 110132.3 | 110296.7 | 1.95E-65 | 0.6957 | 0.5882 |
| lmer_intr_transverse.temporal | 102753.2 | 102900.2 | 0.032413 | 0.9130 | 0.7247 |
| lmer_slope_transverse.temporal | 102750.3 | 102914.7 | 0.032413 | 0.9144 | 0.7258 |
| lmer_intr_insula | 88693.2 | 88840.26 | 7.61E-14 | 0.8263 | 0.4416 |
| lmer_slope_insula | 88636.79 | 88801.14 | 7.61E-14 | 0.8290 | 0.4420 |

