## [Editor Report · eLife assessment]

Duan et al analyzed brain imaging data in UKBK and divided structural brain aging into two groups, revealing that one group is more vulnerable to aging and brain-related diseases compared to the other group. Such subtyping could be **valuable** and utilized in predicting and diagnosing cognitive decline and neurodegenerative brain disorders in the future. This discovery, supported by **solid** evidence, harbors a substantial impacts in aging and brain structure and function.

---

## [Referee Report · Reviewer #1 (Public Review)]

Summary:

Duan et al analyzed brain imaging data in UKBK and found a pattern in brain structure changes by aging. They identified two patterns and found links that can be differentiated by the categorization.

Strengths:

This discovery harbors substantial impacts in aging and brain structure and function.

Weaknesses:

Therefore, the study requires more validation efforts. Most importantly, data underlying the stratification of two groups are not obvious and lack further details. Can they also stratified by different method? i.e. PCA?

Any external data can be used for validation?

Other previous discoveries or claims supporting the results of the study should be explored to support the conclusion.

Sex was merely used as a covariate. Were there sex-differences during brain aging? Sex ratio difference in group 1 and 2?

Although statistically significant, Fig 3 shows minimal differences. LTL and phenoAge is displayed in adjusted values but what is the actual values that differ between pattern 1 and 2?

It is not intuitive to link gene expression result shown in Fig 8 and brain structure and functional differences between pattern 1 and 2. Any overlap of genes identified from analyses shown in Fig 6 (GWAS) and 8 (gene expression)?

---

## [Referee Report · Reviewer #2 (Public Review)]

Summary:

The authors aimed to understand the heterogeneity of brain aging by analyzing brain imaging data. Based on the concept of structural brain aging, they divided participants into two groups based on the volume and rate of decrease of gray matter volume (GMV). The group with rapid brain aging showed accelerated biological aging and cognitive decline and was found to be vulnerable to certain neuropsychiatric disorders. Furthermore, the authors claimed the existence of a "last in, first out" mirroring pattern between brain aging and brain development, which they argued is more pronounced in the group with rapid brain aging. Lastly, the authors identified genetic differences between the two groups and speculated that the cause of rapid brain aging may lie in genetic differences.

Strengths:

The authors supported their claims by analyzing a large amount of data using various statistical techniques. There seems to be no doubt about the quality and quantity of the data. Additionally, they demonstrated their strength in integrating diverse data through various analysis techniques to conclude.

Weaknesses:

The authors provided appropriate answers to the reviewers' questions and revised the manuscript accordingly, and as a result, the paper has been edited to be more easily understood.

---

## [Author Response]

The following is the authors’ response to the original reviews.

**Public Reviews:**

**Reviewer #1 (Public Review):**
Summary:Duan et al analyzed brain imaging data in UKBK and found a pattern in brain structure changes by aging. They identified two patterns and found links that can be differentiated by the categorization.Strengths:This discovery harbors a substantial impact on aging and brain structure and function.Weaknesses:(1) Therefore, the study requires more validation efforts. Most importantly, data underlying the stratification of the two groups are not obvious and lack further details. Can they also stratified by different methods? i.e. PCA?

Response: Thanks for the comment. In this study, principal component analysis (PCA) was applied to individualized deviation of anatomic region of interest (ROI) for dimensionality reduction, which yielded the first 15 principal components explaining approximately 70% of the total variations for identifying longitudinal brain aging patterns. These two patterns can be stratified by both linear and non-linear dimensionality reduction methods: PCA and locally linear embedding (LLE)1. The grey matter volume (GMV) of 40 ROIs at baseline were linearly adjusted for sex, assessment center, handedness, ethnic, intracranial volume (ICV), and second-degree polynomial in age to be consistent with the whole-brain GMV trajectory model. There was a clear boundary between two patterns in the projected coordinate space, indicating distinct structural differences in brain aging between the two patterns (Author response image 1).

**Author response image 1. sa3fig1:** Stratification of the identified brain aging patterns using linear and non-linear dimensionality reduction methods. (a) The principal component space of PC1 and PC2, and (b) two-dimensional projected locally linear embedding space derived from brain volumetric measures. Points have been colored and shaped according to grouping labels of the brain aging patterns.

(2) Are there any external data that can be used for validation?

Response: Thanks for the comment. We were given access to the Alzheimer’s Disease Neuroimaging Initiative (ADNI) study, which aimed at determining the relationships between clinical, cognitive, imaging, genetic, and biochemical biomarkers across the entire spectrum of Alzheimer’s disease. ADNI recruits participants aged between 55 and 90 years at 57 sites in the United States and Canada, who undergo a series of initial tests that are repeated at intervals over subsequent years.

Unfortunately, there are no appropriate and sufficient data, especially clinical, cognitive, and genetic data, to support unbiased validation of the heterogeneity in structural brain aging patterns. Only 890 (31.83%) of the 2796 subjects included in the ADNI were cognitively normal, of which 656 were included in the analyses after quality control of structural MRI and exclusion of missing covariate, with a mean age at the screen visit of 70.8 years (SD = 6.48 years), and 60.21% of the subjects were female. Thus, there are significant differences between ADNI and UK Biobank in terms of the population composition, with ADNI collecting more older subjects due to its focus on defining the progression of Alzheimer’s disease.

Moreover, among 656 subjects with structural imaging data, the dataset used to validate the clinical, cognitive, and genetic manifestations of the brain aging patterns were missing to varying degrees. For example, blood biochemistry tests and telomere length data were missing at baseline by approximately 58% and 82% respectively, and genotype data were not assayed for more than 70 percent of the subjects. As for cognitive function tests, only the results of Mini-Mental State Examination were complete, while other tests such as the Trail Making Test and Digit Span Backward were available for less than 10 percent of subjects.

(3) Other previous discoveries or claims supporting the results of the study should be explored to support the conclusion.

Response: Thanks for the suggestion. As we mentioned in the manuscript lines 274-277, participants with brain aging pattern 2 (lower baseline total GMV and more rapid GMV decrease) were characterized by accelerated biological aging and cognitive decline. Previous research on brainAGE2,3 (the difference between chronological age and the age predicted by the machine learning model of brain imaging data) showed that as a biomarker of accelerated brain aging, people with older brainAGE have accelerated biological aging and early signs of cognitive decline, which is consistent with our discoveries in this study (lines 302-306).

Further, genome-wide association studies identified significant genetic loci contributing to accelerated brain aging, some of which can be found in pervious GWAS on image-derived phenotypes4, such as regional and tissue volume, cortical area and white matter tract measurements, and specific brain aging mode using a data-driven decomposition approach5 (lines 207-213).

In addition, we demonstrated the “last in, first out” mirroring patterns between structural brain aging and brain development, and found that mirroring patterns are predominantly localized to the lateral / medial temporal cortex and the cingulate cortex, noted in the manuscript lines 231-234. Large differences in the patterns of change between adolescent late development and aging in the medial temporal cortex were previously found in studies of brain development and aging patterns6 (lines 315-317).

(4) Sex was merely used as a covariate. Were there sex differences during brain aging? What was the sex ratio difference in groups 1 and 2?

Thanks for the comment. Sex differences during brain aging can be observed by investigating sex-stratified whole-brain GMV trajectories. We fitted the growth curve and estimated rate of change for total grey matter volume (TGMV) separately for male and female using generalized additive mixed effect models (GAMM), which included 40,921 observations from 17,055 males and 19,958 females (Author response image 2). Overall, among healthy participants aged 44-82 years in UK Biobank, males overall had higher total GMV and a faster rate of GMV decrease over time, while females had lower total GMV and a lower rate of GMV decrease. Similar conclusion can be found in normative brain-volume trajectories across the human lifespan7 . Supplementary Table 5 showed baseline and demographic characteristics for all participants and participants stratified by brain aging patterns. There were slightly more females than males among the total participants and for brain aging pattern 1 (53.4%) and pattern 2 (54.4%), and χ^2 tests showed no significant difference in the sex ratio between the two patterns (P = 0.06).

**Author response image 2. sa3fig2:** Total gray matter volume (TGMV) (a) and the estimated rate of change (b) for females (red) and males (blue). Rates of volumetric change for total gray matter and each ROI were estimated using GAMM, which incorporates both cross-sectional between-subject variation and longitudinal withinsubject variation from 22,067 observations for 19,958 females, and 18,854 observations for 17,055 males. Covariates include assessment center, handedness, ethnic, and ICV. Shaded areas around the fit line denotes 95% CI.

(5) Although statistically significant, Figure 3 shows minimal differences. LTL and phenoAge are displayed in adjusted values but what are the actual values that differ between patterns 1 and 2?

Response: Thanks for the comment. We have modified the visualization of Figure 3 in the revised manuscript by adjusting the appropriate axes for leucocyte telomere length (LTL) and PhenoAge variables and removing the whisker from the boxplot. Associations between biological aging biomarkers and brain aging patterns were listed in Supplementary Table 6. Compared to brain aging pattern 1, participants in pattern 2 with more rapid GMV decrease had shorter leucocyte telomere

length (P = 0.009, Cohen’s D = -0.028) and higher PhenoAge (P = 0.019, Cohen’s D = 0.027) without covariate adjustment. Specifically, participants in brain aging pattern 1 had average Z-standardized LTL 0.083 (SD 0.98) and average PhenoAge 41.35 years (SD 8.17 years), and those in pattern 2 had average Z-standardized LTL 0.055 (SD 0.97) and average PhenoAge 41.58 years (SD 8.32 years).

(6) It is not intuitive to link gene expression results shown in Figure 8 and brain structure and functional differences between patterns 1 and 2. Any overlap of genes identified from analyses shown in Figure 6 (GWAS) and 8 (gene expression)?

Response: Thanks for the comment. We apologize for the confusion. As we mentioned in the Result Section Gene expression profiles were associated with delayed brain development and accelerated brain aging, seventeen of the 45 genes mapped to GWAS significant SNP were found in Allen Human Brain Atlas (AHBA) dataset. Gene expression of *LGR4* (rspearman = 0.56, Ppermutation = 2.5 × 10-4) were significantly associated with delayed brain development, and ESR1 (rspearman = 0.53, Ppermutation = 1.5 × 10-4) and *FAM3C* (rspearman = -0.37, Ppermutation = 0.004) were significantly associated with accelerated brain aging. *BDNF-AS* was positively associated with both delayed brain development and accelerated brain aging after spatial permutation test. Full association between gene expression profiles of mapped genes and estimated APC during brain development / aging were presented in Supplementary Tables 12 and 13, respectively.

Furthermore, we screened the genes based on their contributions and effect directions to the first PLS components in brain development and brain aging. We have found genes mapped to GWAS significant SNP among the genes screened for inclusion in the functional enrichment analysis (Author response table 1), with *LGR4* (PLSw1(*LGR4*) = 3.70, P.FDR = 0.002) associated with delayed development and ESR1 (PLSw1(*ESR1*) = 3.91, P.FDR = 6.12 × 10-4) and FAM3C (PLSw1(*FAM3C*) = -3.68, P.FDR = 0.001) associated with accelerated aging.

**Author response table 1. sa3table1:** Contributions and effect directions of the first PLS components in brain development and brain aging of genes that mapped to GWAS significant SNP. The bold P values reflect significance (P < 0.005, inclusion in the functional enrichment analysis) after FDR correction.

IMAGEN			
Gene	bootstrap.weights	pvalue	p.adjust
LGR4	3.70	2.17E-04	2.00E-03
CCDC170	1.59	1.11E-01	2.41E-01
SUFU	1.35	1.77E-01	3.31E-01
SPPL2C	0.97	3.32E-01	5.04E-01
ACTRIA	0.94	3.49E-01	5.21E-01
MAPT	0.89	3.75E-01	5.46E-01
KANSL1-AS1	0.76	4.50E-01	6.12E-01
ARHGAP27	0.67	5.03E-01	6.58E-01
LIN7C	0.52	6.03E-01	7.39E-01
ARL17B	0.45	6.53E-01	7.76E-01
ESRI	0.16	8.75E-01	9.29E-01
BDNF-AS	0.00	9.99E-01	9.99E-01
FAM3C	-0.10	9.20E-01	9.57E-01
STH	-1.27	2.06E-01	3.68E-01
PLEKHM1	-2.39	1.70E-02	6.13E-02
KANSLI	-2.40	1.65E-02	5.99E-02
LRRC37A4P	-2.69	7.09E-03	3.13E-02
UKBiobank			
Gene	bootstrap.weights	pvalue	p.adjust
BDNF-AS	4.73	2.25E-06	5.11E-05
ARHGAP27	4.02	5.92E-05	4.38E-04
ESRI	3.91	9.31E-05	6.13E-04
SPPL2C	3.30	9.56E-04	3.69E-03
SUFU	2.20	2.79E-02	5.72E-02
LIN7C	2.15	3.14E-02	6.33E-02
CCDC170	2.08	3.78E-02	7.34E-02
LGR4	1.76	7.88E-02	1.35E-01
ARL17B	1.48	1.39E-01	2.13E-01
PLEKHM1	0.69	4.93E-01	5.89E-01
LRRC37A4P	0.27	7.84E-01	8.39E-01
KANSL1-AS1	0.17	8.63E-01	9.00E-01
KANSLI	-0.56	5.75E-01	6.64E-01
STH	-1.03	3.04E-01	4.01E-01
MAPT	-1.38	1.67E-01	2.47E-01
ACTRIA	-2.13	3.35E-02	6.67E-02
FAM3C	-3.68	2.38E-04	1.23E-03

**Reviewer #2 (Public Review):**
Summary:The authors aimed to understand the heterogeneity of brain aging by analyzing brain imaging data. Based on the concept of structural brain aging, they divided participants into two groups based on the volume and rate of decrease of gray matter volume (GMV). The group with rapid brain aging showed accelerated biological aging and cognitive decline and was found to be vulnerable to certain neuropsychiatric disorders. Furthermore, the authors claimed the existence of a "last in, first out" mirroring pattern between brain aging and brain development, which they argued is more pronounced in the group with rapid brain aging. Lastly, the authors identified genetic differences between the two groups and speculated that the cause of rapid brain aging may lie in genetic differences.Strengths:The authors supported their claims by analyzing a large amount of data using various statistical techniques. There seems to be no doubt about the quality and quantity of the data. Additionally, they demonstrated their strength in integrating diverse data through various analysis techniques to conclude.Weaknesses:There appears to be a lack of connection between the analysis results and their claims. Readers lacking sufficient background knowledge of the brain may find it difficult to understand the paper. It would be beneficial to modify the figures and writing to make the authors' claims clearer to readers. Furthermore, the paper gives an overall impression of being less polished in terms of abbreviations, figure numbering, etc. These aspects should be revised to make the paper easier for readers to understand.
**Recommendations for the authors:**

**Reviewer #1 (Recommendations For The Authors):**
Gray matter volume (GMV) is defined later in the manuscript and may confuse readers.

Response: Thanks for the comment. We have now defined GMV upon its first appearance in the manuscript.

**Reviewer #2 (Recommendations For The Authors):**
(1) In conducting GWAS, the authors used total GMV at the age of 60 as a phenotype (line 195). It would be beneficial to provide additional explanation as to why only the data from individuals aged 60 were utilized, especially considering the ample availability of GMV data.

Response: Thanks for the comment and we apologize for the confusion. As we mentioned in the Methods Section Genome Wide Association Study to identify SNPs associated with brain aging patterns, we performed Genome-wide association studies (GWAS) on individual deviations of total GMV relative to the population average at 60 years using PLINK 2.0. Therefore, data from all individuals were used in the GWAS, rather than only those aged at 60y. To accomplish this, deviation of total GMV from the population average for each participant at age 60y was calculated using mixed effect regression model as described in the Methods Section Identification of longitudinal brain aging patterns.

(2) Whole-brain gene expression data was linked to GMV (Line 237). Gray matter is known to account for about 40% of the total brain. Thus, interpreting whole-brain data in connection with GMV might introduce significant errors. Could this potential source of error be addressed?

Response: Thanks for the comment. In our study, the Allen Human Brain Atlas (AHBA) dataset were processed using abagen toolbox version 0.1.3 (https://doi.org/10.5281/zenodo.5129257) with Desikan-Killiany atlas8, resulting in a matrix (83 regions × 15,633 gene expression levels) of transcriptional level values that contains brain structure of cortex and subcortex in bilateral hemispheres, and brainstem. Only data from 34 cerebral cortex regions, but not the whole brain, were included in the analysis of the association between regional change rate of gray matter volume and gene expression profiles using partial least squares (PLS) regression. We have clarified in the revised manuscript that we utilized AHBA microarray expression data from regions of interest (ROIs) in the cortex.

(3) The paper lacks biological interpretation of the important genetic factors (SNPs and genes) for brain aging discovered in this study, as well as the results of gene ontology analysis. Many readers would be curious about the biological significance of these genetic differences and what kind of outcomes they may produce.

Response: Thanks for the suggestion. As we mentioned in our manuscript, six independent single nucleotide polymorphisms (SNPs) were identified at genome-wide significance level (P < 5 ×1 0-8) (Fig. 6). Among them, two SNPs (rs10835187 and rs779233904) were also found to be associated with multiple brain imaging phenotypes in previous studies, such as regional and tissue volume, cortical area and white matter tract measurements. Compared to the GWAS using global gray matter volume as the phenotype, our GWAS revealed additional signal in chromosome 7 (rs7776725), which was mapped to the intron of FAM3C and encodes a secreted protein involved in pancreatic cancer and Alzheimer's disease. This signal was further validated to be associated with specific brain aging mode by another study using a data-driven decomposition approach. In addition, another significant locus (rs10835187, P = 1.11 ×1 0-13) is an intergenic variant between gene LGR4-AS1 and LIN7C, and was reported to be associated with bone density, and brain volume and total cortical area measurements. LIN7C encodes the Lin-7C protein, which is involved in the localization and stabilization of ion channels in polarized cells, such as neurons and epithelial cell. Previous study has revealed the association of both allelic and haplotypic variations in the LIN7C gene with ADHD. In addition, ESR1 was found to be involved in I-kappaB kinase/NF-kappaB signaling in the functional enrichment associated with accelerated brain aging (Figure 8 and Supplementary Figure 5), and its activation leads to a variety of human pathologies such as neurodegenerative, inflammatory, autoimmune and cancerous disease9.

In summary, the analyses from using the databases of GO biological processes and KEGG Pathways indicate synaptic transmission as an important process in the common mechanisms of brain development and aging, and cellular processes (autophagy), as well as the progression of neurodegenerative diseases, are important processes in the mechanisms of brain aging.

(4) As mentioned in the public review, it would be helpful if figures were revised to more clearly represent the claims.(4.1) For Figure 1, it would be beneficial to explain how the authors analyzed the differences between the mentioned cross-section and longitudinal trajectory, which they identified as a strength of the study.

Response: We have added the strengths of adopting longitudinal data for modeling brain aging trajectories compared to only using cross-sectional data in Figure 1 caption in the revised manuscript:

“Fig. 1 Overview of the study workflow. a, Population cohorts (UK Biobank and IMAGEN) and data sources (brain imaging, biological aging biomarkers, cognitive functions, genomic data) involved in this study. b, Brain aging patterns were identified using longitudinal trajectories of the whole brain GMV, which enabled the capturing of long-term and individualized variations compared to only use cross-sectional data, and associations between brain aging patterns and other measurements (biological aging, cognitive functions and PRS of major neuropsychiatric disorders) were investigated. c, Mirroring patterns between brain aging and brain development was investigated using ztransformed brain volumetric change map and gene expression analysis.”

(4.2) In Figure 3, it's challenging to distinguish differences between patterns 1 and 2 in LTL and PhenoAge. (e.g. It's unclear whether Pattern 1 is higher or lower). Clarifying this visually would be useful.

Response: We have modified the visualization of Figure 3 in the revised manuscript by adjusting the appropriate axes for leucocyte telomere length (LTL) and PhenoAge variables and removing the whisker from the boxplot.

**Author response image 3. sa3fig3:** Distributions of biological aging biomarkers (leucocyte telomere length (LTL) and PhenoAge) among participants with brain aging patterns 1 and 2.

(4.3) Figure 7 explains the mirroring pattern, but it's hard to discern significant differences from the figures alone (especially in Figures 7b and 7c). Using an alternative method (graph, etc.) to clearly represent this would be appreciated.

Response: We have included an arrow pointing to the brain regions with significant differences in each subfigure.

**Author response image 4. sa3fig4:** The “last in, first out” mirroring patterns between brain development and brain aging.

(5) Abbreviations should be explained when they are first introduced in the paper. For example, GMV continues to be used without explanation, and in line 203, it is written out as 'gray matter volume'. ADHD and ASD first appear at line 172, but the explanation is found in lines 177-178. Additionally, there are terms without explanations in the manuscript. For instance, BMI is not explained in the main manuscript but is defined in the Supplementary Information (Table S6).

Response: We have corrected the inappropriate formatting regarding misplaced and missing abbreviations in the revised manuscript and Supplementary Information.

(6) Figure numbers should follow the order of appearance in the paper. The first Supplementary Fig. in the manuscript is Supplementary Figure 3. It should be Supplementary Figure 1.

Response: We have relabeled the figures with the order of appearance in the paper in the revised manuscript and Supplementary Information.

Reference:

(1) Roweis, S. T. & Saul, L. K. Nonlinear dimensionality reduction by locally linear embedding. *science* 290, 2323–2326 (2000).

(2) Christman, S. *et al.* Accelerated brain aging predicts impaired cognitive performance and greater disability in geriatric but not midlife adult depression. *Translational Psychiatry* 10, 317 (2020).

(3) Elliott, M. L. *et al.* Brain-age in midlife is associated with accelerated biological aging and cognitive decline in a longitudinal birth cohort. *Molecular psychiatry* 26, 3829–3838 (2021).

(4) Smith, S. M. *et al.* An expanded set of genome-wide association studies of brain imaging phenotypes in UK Biobank. *Nature neuroscience* 24, 737–745 (2021).

(5) Smith, S. M. *et al.* Brain aging comprises many modes of structural and functional change with distinct genetic and biophysical associations. *elife* 9, e52677 (2020).

(6) Tamnes, C. K. *et al.* Brain development and aging: overlapping and unique patterns of change. *Neuroimage* 68, 63–74 (2013).

(7) Bethlehem, R. A. *et al.* Brain charts for the human lifespan. *Nature* 604, 525–533 (2022).

(8) Desikan, R. S. *et al.* An automated labeling system for subdividing the human cerebral cortex on MRI scans into gyral based regions of interest. *Neuroimage* 31, 968–980 (2006).

(9) Singh, S. & Singh, T. G. Role of nuclear factor kappa B (NF-κB) signalling in neurodegenerative diseases: an mechanistic approach. *Current Neuropharmacology* 18, 918–935 (2020).